# The *Escherichia coli* chromosome moves to the replisome

Konrad Gras[1], David Fange [1]✉ & Johan Elf [1]✉

In *Escherichia coli*, it is debated whether the two replisomes move independently along the two chromosome arms during replication or if they remain spatially confined. Here, we use high-throughput fluorescence microscopy to simultaneously determine the location and short-time-scale (1 s) movement of the replisome and a chromosomal locus throughout the cell cycle. The assay is performed for several loci. We find that (*i*) the two replisomes are confined to a region of ~250 nm and ~120 nm along the cell's long and short axis, respectively, (*ii*) the chromosomal loci move to and through this region sequentially based on their distance from the origin of replication, and (*iii*) when a locus is being replicated, its short time-scale movement slows down. This behavior is the same at different growth rates. In conclusion, our data supports a model with DNA moving towards spatially confined replisomes at replication.

Bacterial chromosome organization is influenced by various essential processes, including DNA replication, segregation, and gene expression. In *Escherichia coli*, the chromosome is compacted, with coiled loops extending from an axial core[1,2], forming a bottle-brush-like structure[3] along the long axis of the cell. While being highly coiled to fit into the bacterial cell, chromosomal DNA has to be replicated and segregated before cell division to ensure propagation. The protein complex responsible for replicating DNA, the replisome[4,5], initiates replication at the origin of replication, *oriC*, and replicates the circular chromosome sequence bidirectionally towards the terminus region, *ter*. Contrary to the consensus view of the order in which the DNA sequence of the *E. coli* chromosome is replicated, the spatial aspects of this process have been debated for almost 30 years. The starting point is usually set to 1998, when Lemon & Grossman[6] observed that GFP-labeled subunits of the replisome in *Bacillus subtilis* were found in well-defined regions rather than shifting positions throughout the cell cycle. Based on this, they suggested that the bacterial chromosome is moving through the spatially confined replisomes while it is being replicated. Similar observations of spatially confined replisomes were later made in other bacteria using different replisome markers[7,8]. However, reports of rapidly segregating replisomes were also presented[9,10].

During the same time period, large efforts were made to characterize the *E. coli* chromosome dynamics. Initially, this primarily relied on FISH to different loci[11-13] in fixed cells. Later, methods using the binding of fluorescent proteins to specific DNA sequences were developed[14,15], which also allowed for the tracking of loci in living cells. The *ter* and *oriC* regions were mainly found at opposite cell poles[12]. The right and left arms were either found to be on opposite sides of the cell middle[16,17], or with the two chromosome arms aligned next to each other from pole to pole[18], possibly depending on the growth conditions[19,20]. Following initiation of replication, the two copies of the *oriC* locus were found to be spatially colocalized for extended periods[21], a phenomenon coined sister chromosome cohesion. Cohesion was found to be dependent on Topoisomerase IV[22] and was also observed for other loci[12]. The *ter* locus was found to make rapid transitions between the old and new poles, depending on a set of pole-bound proteins, namely MatP, ZapA, ZapB, and FtsK[23]. The *ter* transition was also shown to be dependent on active replication[23]. Using protein-based locus markers, Javer et al.[24] characterized loci movement on the time scale of seconds and found that *ter* has a minimum in short-time-scale movement.

Using FISH-based loci localization together with immuno-fluorescently labeled replisome markers, Bates & Kleckner[12] made observations that contradict the idea of spatially confined replisomes[6]. In line with refs. 9,10, Bates & Kleckner[12]. instead found that replisomes separate after initiation, suggesting that it is the replisomes that are moving on more stationary chromosomes. This finding was corroborated by Reyes-Lamothe et al.[25] who combined their newly developed locus-labeling method[15] with fluorescently labeled single-stranded

[1]Dept. of Cell and Molecular Biology, Science for Life Laboratory, Uppsala University, Uppsala, Sweden. ✉e-mail: david.fange@icm.uu.se; johan.elf@icm.uu.se

binding protein (SSB). This study, where living *E. coli* cells carrying markers on chromosomal loci and replisomes in the same cells were imaged in a time-lapse manner, suggested that the replisomes move towards more stationary chromosome loci.

The view on the dynamics of the replisomes then shifted again with three publications[18,26,27] favoring Lemon & Grossman's[6] suggestion of chromosomes moving through spatially confined replisomes. Using snapshot-based imaging with the *parS*/ParB system, Youngren et al.[18] found that chromosome loci split at well-defined intracellular regions. Using a strain carrying the replication fork marker SSB, they found that the replisomes form foci in the same regions where the loci split. These observations were corroborated by Cass et al.[26] who used time-lapse microscopy to track chromosome loci in single living cells over time. Also here, the chromosome loci move into a confined region in which loci splitting occurs. By tracking fluorescently labeled replisome sub-units (DnaN), Mangiameli et al.[27] showed that also single replisomes stay confined within the region in which loci are expected to split. The two sister replisomes were detected either as a single fluorescent focus or occasionally split apart far enough to be observed as two fluorescent foci. Regions with confined replisomes with occasional splitting and subsequent merging of the replisome foci were also observed in refs. [28,29].

The latest publication reporting the results of simultaneous tracking of replisomes and chromosome imaging changes the perspective yet again. Using mutants with increased cell width, Japaridze et al.[30] showed that, following initiation, the two sister replisomes split into two fluorescent foci, which moved along the chromosome during replication.

Knowing where replication occurs is crucial for understanding the growth-dependent organization of the *E. coli* chromosome. However, reconciling the different observations reviewed above is non-trivial. The most direct observation of time-lapsed loci-replisome interdependent movement in wild-type-shaped cells was made by Reyes-Lamothe et al.[25], with the conclusion that the replisomes trail along the DNA during replication. This was later backed up by Japaridze et al.[30] although in a cell shape very different from the wild type. At the same time, there are multiple observations that replisomes are spatially confined[27–29], and chromosome loci studies[18,26] suggest a confined region of replication. These findings are, however, not based on the same set of cells, and differences in growth conditions between different publications make direct comparisons difficult.

Here we revisited the question of how the localization of a locus is affected by replication. To do so, we performed experiments that combined (i) high-throughput cell cycle-scale time-lapse fluorescence microscopy of live *E. coli* cells kept in steady-state growth over long time scales[29] with (ii) dual labeling of replication and chromosome loci in the same cell[25] and (iii) short-time-scale imaging[24]. This way, we captured the movements of replisomes and chromosome loci on a short-time scale (seconds) as well as the cell-cycle time scale and measured replisome-loci distances in single cells. Since the cells were kept in steady-state growth over long periods, we could follow each cell from division to division to ensure that the distributions of locations, short-time-scale displacements, and distances were based on the same set of cells.

Initially, we focused our efforts on the origin of replication, *oriC*, which by physical necessity colocalizes with the replisome at replication initiation. We measured the short-time-scale movement of *oriC* over the cell cycle and observed a decrease at replication initiation. A similar behavior was observed for loci further away from *oriC*; the decrease in short-time-scale movement occurred when the loci were colocalized with the confined replisome. Lastly, we show that the observed behavior is consistent at different growth rates. Our results strongly suggest that the chromosome moves in response to being replicated while the replisomes stay in a confined intracellular region during replication.

## Results

### An *oriC*-proximal chromosome locus shows more dynamics over the cell cycle as compared to the replisome

To investigate if the chromosome or the replisome, or both, relocate during the replication process, we constructed a strain with fluorescent labels of different colors on the replisome and an *oriC*-proximal chromosome locus. Initially, we focused on *oriC* since initiation occurs within a narrow range of cell sizes[29], making interpretation more straightforward. We constructed an *E. coli* MG1655 strain with DnaN translationally fused with mCherry[31] and a YFP-based fluorescent repressor-operator system (FROS)[15] 34 kb from *oriC*. The FROS-based locus-labeling system included an array of 12 *malO* operators and constitutive expression of the gene encoding MalI-SYFP2. We grew the strain in a mother-machine-type microfluidic chip[32], where we imaged the bacteria every minute over several hours. Cell outlines were estimated by segmentation of phase-contrast microscopy images and assembled into tracks of cell lineages. Fluorescence images of the replisome and *oriC* labels were acquired back-to-back (two 25-ms laser excitation pulses of different colors, <1 ms apart), every minute. The emitted fluorescence was split into different parts of a single camera chip. Replication initiation events were identified by tracking replisome foci positions using u-track[33] to find the start of a replisome trajectory[29].

To visualize where the replisome and the *oriC*-proximal loci are positioned within the cells, each replisome and locus fluorescent focus was placed in a cell internal coordinate system defined by the cell outline captured using phase-contrast. The area enclosed by the cell outline is hereafter referred to as the cell area. The cells were binned based on cell area, and for each bin, we generated bi-variate distributions of the long- and short-axis intracellular positions of replisomes and *oriC*-proximal loci (Fig. 1).

At a cell area of ~2 μm², two replisome distributions spawned around the ¼ and ¾ long-axis positions (Fig. 1, top). The appearance of new replisome foci in these positions has previously been shown to represent replication initiation[18,27,28]. Thus, each of the replisome distributions included the locations of a pair of replication forks replicating the chromosomal DNA sequence bidirectionally from *oriC*[25,28]. As expected, the average of the location distribution of the *oriC*-proximal chromosome loci colocalized with the replisome distribution in the size span where initiation occurs (Fig. 1, bottom). During one generation, the replicated *oriCs* must segregate into the daughter cells. We found that the segregation of *oriC* spanned most of the time between two initiations. The replisome distributions, on the other hand, stayed localized at a similar average distance from the middle of the cell throughout the replication process. Note that after cell division, the old cell middle equals the new cell pole (Fig. 1).

Although there were two replication forks in each of the two unimodal replisome distributions, the width, defined as the standard deviation of the location distributions, remained relatively constant along the cell-long-axis, while the width along the short axis was always smaller than that of the long-axis direction and decreased following initiation (Fig. 1, Figs. S1, S3b). This shows that the two replisomes moving bidirectionally on the chromosome stay within a volume smaller than ~125 × 125 × 200 nm ellipsoid (Figs. S1, S3b) throughout the replication process[27]. We will refer to this as the replisome region.

The size of the replisome region as seen in Fig. 1 could arise either as a result of stationary replisomes with cell-to-cell variability in replisome localization (Fig. 2a, right), or from replisomes moving around in the region (Fig. 2a, left), either individually or as pairs of bidirectionally moving replication forks. To discriminate between these two scenarios, we tracked the replisomes in individual cells for 20 minutes following the initiation event and recorded how the root mean squared displacement (RMSD) changed in time (Fig. 2b). Since the replisome distributions showed a minor net average movement away from the cell middle on the time-scale of 20 min (Fig. S4), this movement was subtracted before estimating the RMSD. We used

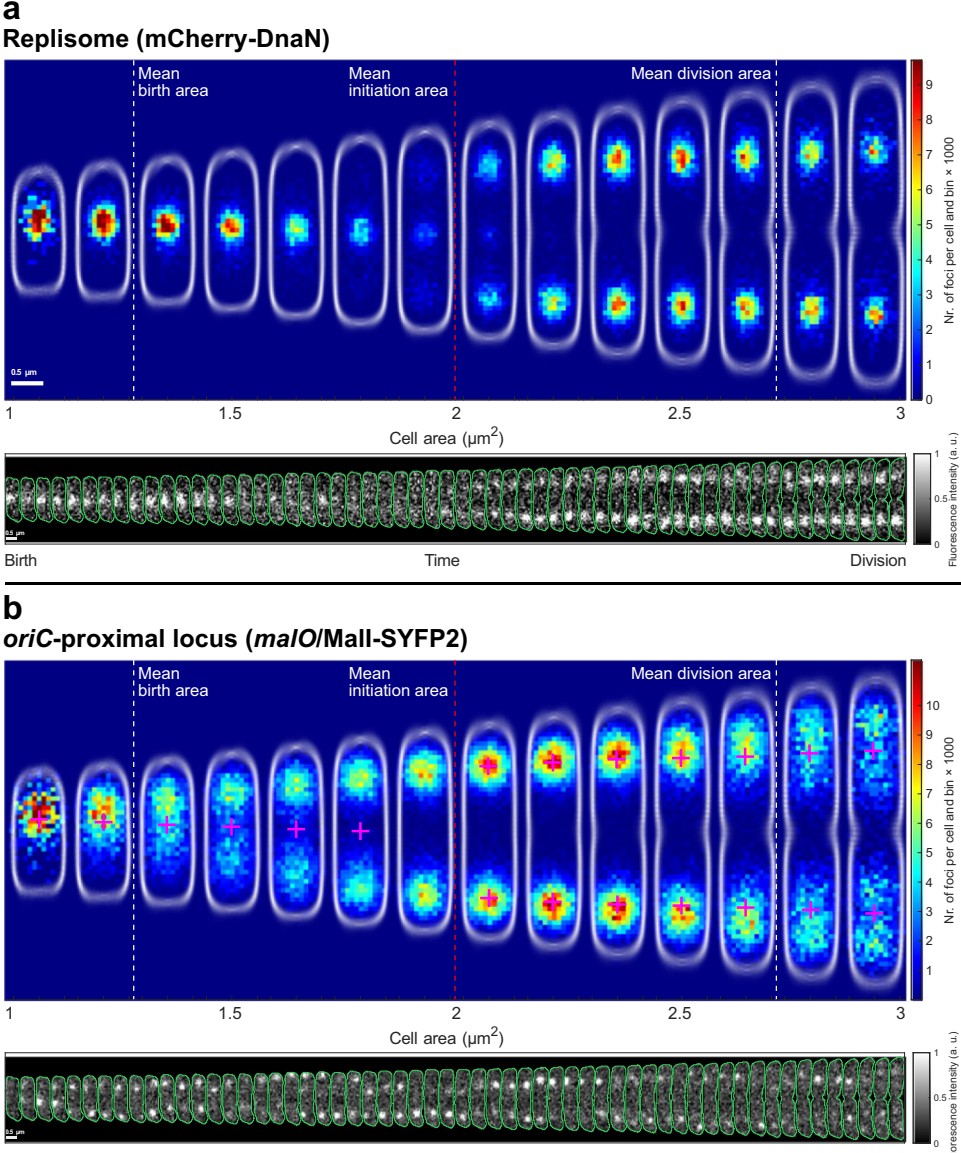

**Fig. 1 | *OriC* and replisome location distributions over the cell cycle.** Two-dimensional histograms of fluorescent foci positions along the long and short cell axis from an *E. coli* strain that simultaneously carries a replisome label (**a** mCherry-DnaN) and an *oriC*-proximal chromosomal locus label (**b** *malO*/MalI-YFP system with an array of operator sequences introduced 34 kb from *oriC*). Below each two-dimensional histogram is an example of a cell with the replisome and *oriC* labels tracked over one generation, where each image of the cell is separated by 1 minute. The color in each histogram heat map indicates the number of foci per cell and histogram bin, where all the two-dimensional histograms use the same bin size and bin positions. The distributions of cell outlines detected in phase-contrast are shown in white. The old cell pole is always pointing up. White dashed lines indicate average birth and division sizes. Red dashed lines indicate the average replication initiation size. The average birth, division, and replication initiation sizes have been rounded to the nearest area bin edge. Magenta plus signs correspond to the mean position of the replisome distributions in the top panel. Statistics of the number of cells and fluorescent foci can be found in Supplementary Dataset 2.

DnaN-YPet[29] as a replisome label for the RMSD estimation, due to its higher signal-to-background ratio compared to the mCherry-based label. The RMSDs plateau at ~0.25 μm in the cell-long-axis direction and ~0.12 μm in the cell short-axis direction. The plateau values for both the short and long axis of the cell were similar to the expected average distance between two random positions in the replisome location distributions, suggesting that the dominating contributor to the width of the replisome region shown in Fig. 1 is replisome movements in individual cells and not cell-to-cell variation. RMSDs for the *oriC*-proximal locus were larger as compared to the RMSDs for the replisome (Fig. 2c) as expected from the larger width of the location distributions in Fig. 1.

Taken together, we find that the average position of the location distribution of the replisome moves far less over the cell cycle as compared to the average position of the location distribution of the *oriC*-proximal locus. Additionally, we find that the width of the location distribution is smaller for the replisome distribution than for the *oriC*-proximal locus at all stages of the cell cycle. These localization patterns are indicative of confined replisomes and unrestrained chromosomal loci. However, they are inconclusive in terms of movements immediately before and after the initiation event, since small changes in the localization of the *oriC*-proximal locus are easily lost in the thermally induced short-time-scale movements of both the loci and the replisomes.

## a

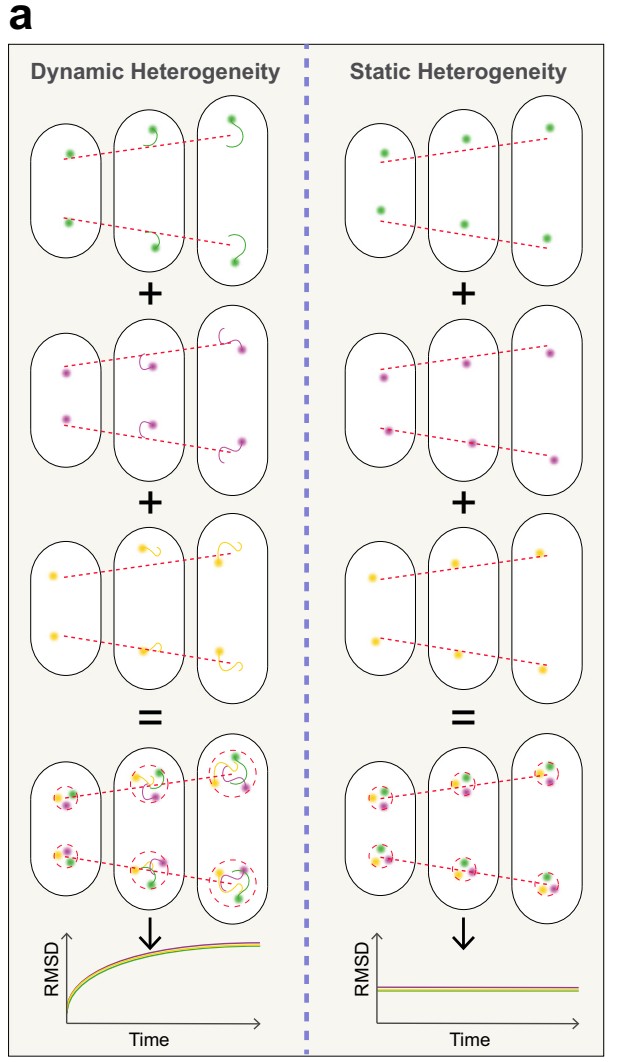

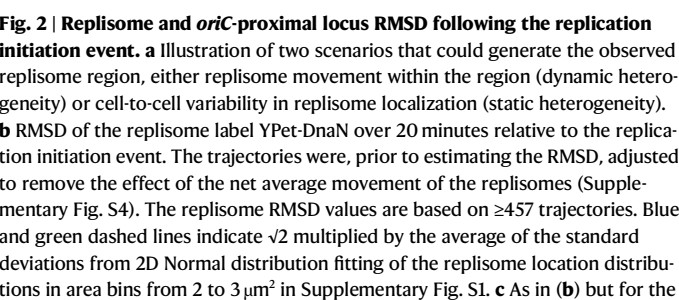

## b

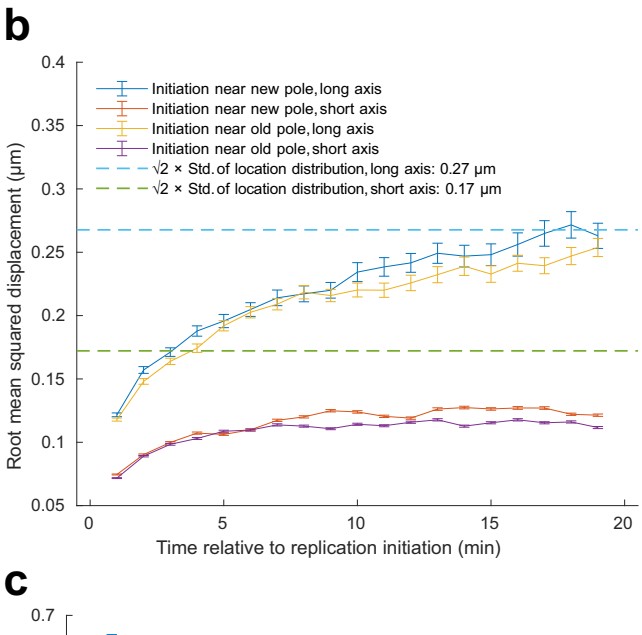

## c

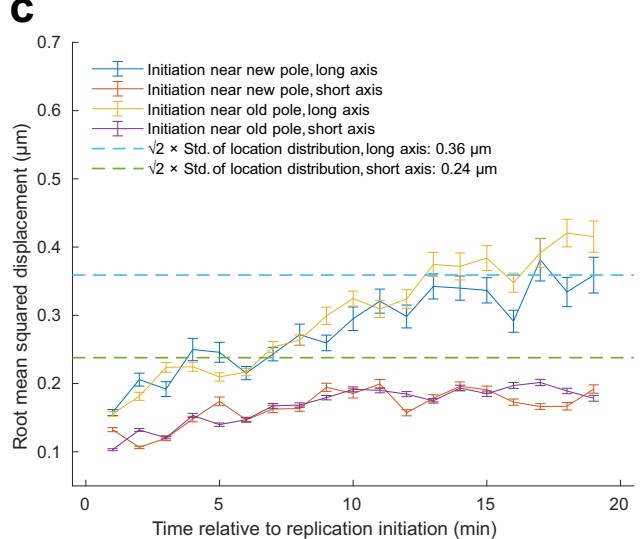

**Fig. 2 | Replisome and *oriC*-proximal locus RMSD following the replication initiation event. a** Illustration of two scenarios that could generate the observed replisome region, either replisome movement within the region (dynamic heterogeneity) or cell-to-cell variability in replisome localization (static heterogeneity). **b** RMSD of the replisome label YPet-DnaN over 20 minutes relative to the replication initiation event. The trajectories were, prior to estimating the RMSD, adjusted to remove the effect of the net average movement of the replisomes (Supplementary Fig. S4). The replisome RMSD values are based on ≥457 trajectories. Blue and green dashed lines indicate √2 multiplied by the average of the standard deviations from 2D Normal distribution fitting of the replisome location distributions in area bins from 2 to 3 μm² in Supplementary Fig. S1. **c** As in (**b**) but for the *oriC*-proximal locus over 20 minutes relative to the replication initiation event based on tracking of the mCherry-DnaN label in the same cells. The trajectories were, prior to estimating the RMSD, adjusted to remove the effect of the net average movement of the *oriC*-proximal locus. The *oriC*-proximal locus RMSD values are based on ≥ 20 trajectories. The blue and green dashed lines are based on 2D Normal distribution fitting of the *oriC* location distributions in area bins from 2 to 3 μm² in Supplementary Fig. S2. Error bars correspond to standard error of the mean. The experiment was performed twice. Replisome RMSD for a repeated experiment is shown in Supplementary Fig. S3. Statistics can be found in Supplementary Dataset 1.

### Replication of a locus coincides with a decrease in its short-time-scale movement

To further characterize the short-time-scale movement of the *oriC*-proximal locus and the replisome, we extended the time-lapse imaging used in Fig. 1 to include two *oriC*-replisome image pairs, acquired 1 second apart. The foci were tracked between the two frames and the 1-second displacements were estimated for both the *oriC*-proximal locus and the replisome. The displacements were binned based on cell size. The distributions of frame-to-frame displacement for each cell area bin for both *oriC* and replisomes are shown in Fig. 3a, b. Included in the figure panels are also the average cell sizes at initiation (red dashed lines) calculated as in Fig. 1. The movement that we measured as 1-second displacements will henceforth be referred to as short-time-scale

movement. We found that the short-time-scale movement of the *oriC*-proximal locus decreased in the cell size window where initiation occurs (Fig. 3a, c). The average short-time-scale movement of the replisome, on the other hand, stayed constant throughout the cell cycle at a level that is, on average lower than that of *oriC* (Fig. 3b, c). Also, the difference in average short-time-scale movements between *oriC* and replisome was slightly larger due to the higher localization uncertainty for the replisome marker mCherry-DnaN as compared to the replisome marker YPet-DnaN. See Fig. S5 for a comparison between mCherry-DnaN (used in Fig. 3) and YPet-DnaN. Measurements of two-dimensional (2D) Euclidean distances between replisomes and *oriC*-proximal loci in the same cells for different cell areas (Fig. 3d) also showed that the displacement minimum coincides with the minimal replisome-*oriC* distance.

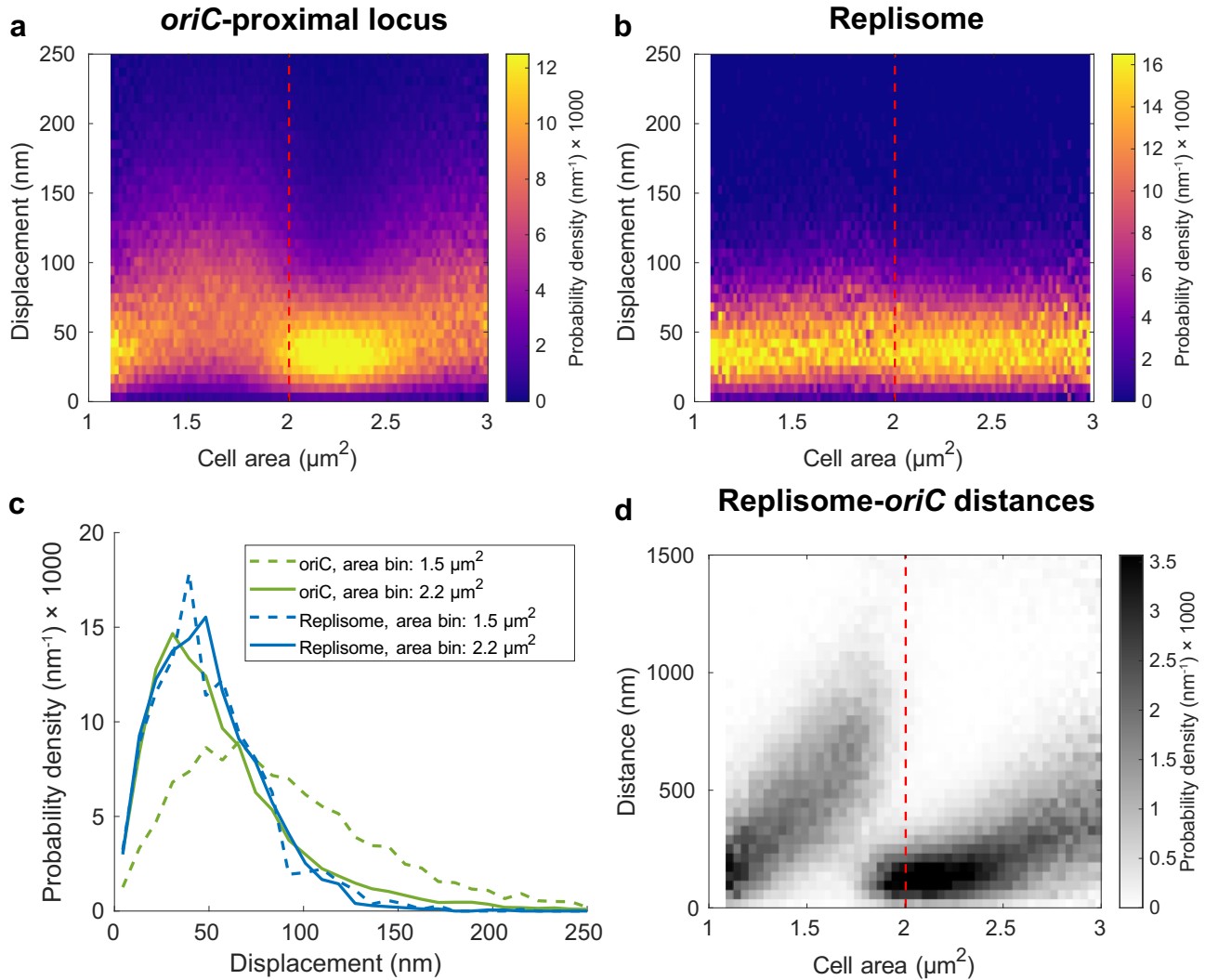

**Fig. 3 | *OriC* and replisome displacements at different cell areas. a**, **b** Frame-to-frame displacement of tracked fluorescent foci between two frames acquired 1 s apart. Foci tracking was performed using the u-track[33] algorithm. Displacements at different cell sizes estimated for (**a**) *oriC*-proximal chromosome loci and (**b**) replisomes. Heat map color shows the probability density for different displacements in each area bin. **c** Displacement distributions from (**a**, **b**) at specific cell sizes. **d** 2D distances between the *oriC*-proximal locus and replisomes for different cell areas. Heat map grayscale shows the probability density for different distances in each area bin. **a**, **b**, **d**: Red dashed lines indicate average replication initiation size. The experiment was performed twice. Statistics can be found in Supplementary Dataset 1.

If the decrease in *oriC* displacements and the replisome-*oriC* distance minimum are due to the replication fork passing the locus, we should see a corresponding pattern for other chromosomal loci at well-defined times after initiation. To test this, we repeated the experiment for 11 strains with the YFP-based FROS labels at different distances from *oriC*. In all 11 strains, we observed displacement minima similar to that of the *oriC*-proximal locus at different cell sizes (Fig. 4). The minima of loci displacements occurred at the same cell area as the minima in replisome-locus distance. The timing of these events was consistent with a model where initiation occurs at a cell size of 2.05 μm², replication takes 45 minutes to complete and is progressing at a constant rate, and the cells are exponentially growing with a doubling time of 50 minutes (see Methods for details). These results show that as loci are replicated, their short-time-scale movement decreases to a similar level as that of the replisome (Fig. 4).

**Replication induces transient spatial repositioning of chromosomal loci**

Using the 12 strains with FROS labels in different positions, we also quantified the short-time-scale movement of chromosomal loci at different spatial positions inside each cell (Fig. 5). We note that minima occur where the location distributions of each locus and the replisome overlap, in agreement with the distance measurements in Fig. 4. For the *oriC*-proximal loci, we find an overall expected behavior, with a decrease in displacement at the cell size interval where initiation occurs (Fig. 3), but the short-time-scale movement does not seem to depend on the position of the locus in the cell's coordinate system. However, for chromosomal loci L4 and R4, an interesting behavior emerged (L4 from Fig. 5b is highlighted in Fig. 5a). At cell sizes of ~1.6 μm², these loci relocate from the new pole to the replisome region, where the loci's short-time-scale movement becomes slower. Shortly after replication, the loci are segregated into their new positions and remain there until the next replication round. As the loci segregate, their short-time-scale movement returns to the pre-replication level. Examples of transient relocalization in single cells are shown in Fig. S11. These observations are consistent with the model in which the chromosome loci move towards a region of spatially confined replisomes, in which the loci replicate. This is then also inconsistent with the model where the replisomes move to the chromosome loci before their replication.

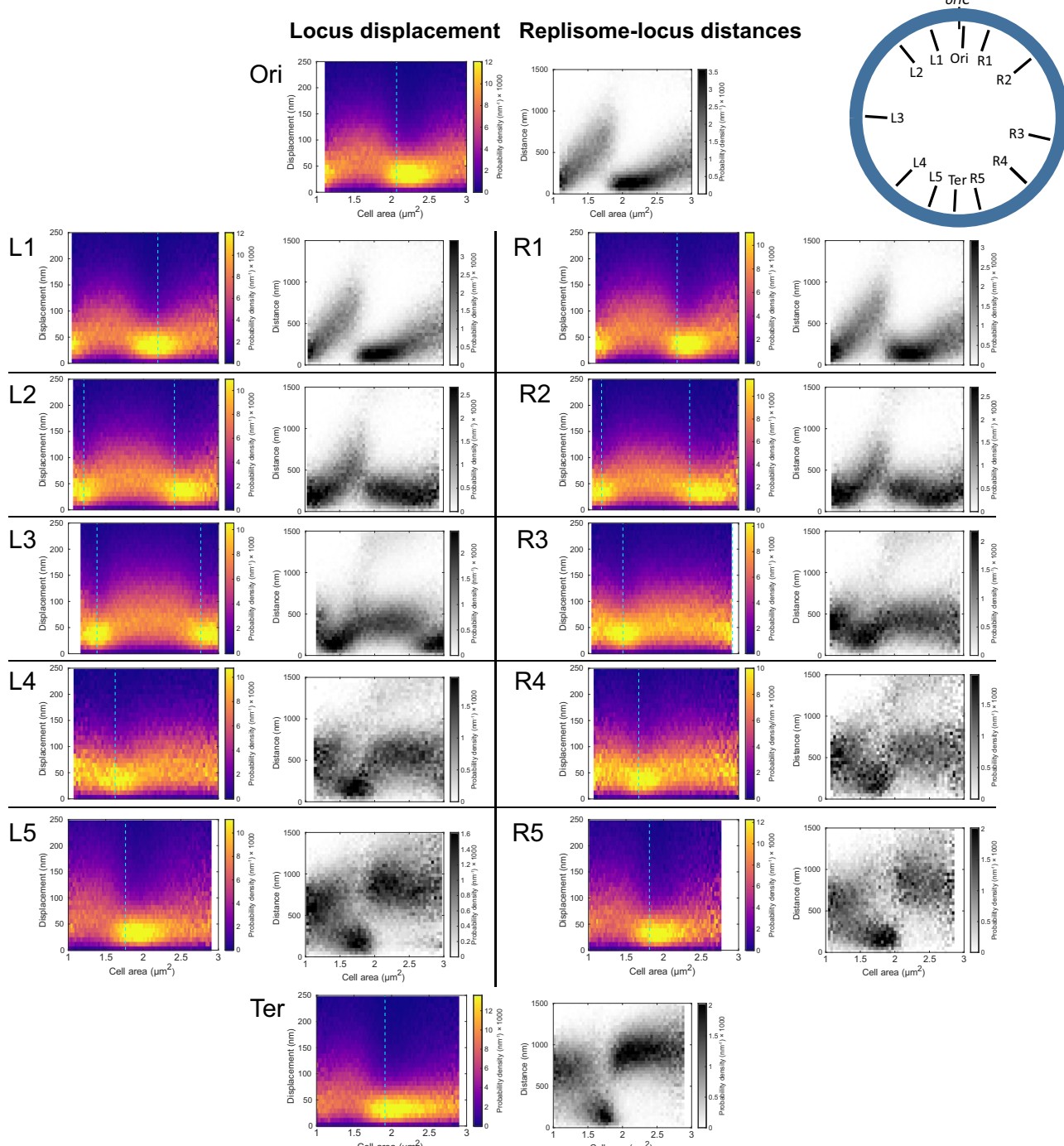

**Fig. 4 | Locus displacement minima coincide with replisome-locus colocalization.** As Fig. 3a (locus displacement) and 3d (replisome-locus distance) but for chromosome locus labels at different chromosomal positions relative to *oriC* (L1-L4, R1-R4, and Ter). Colors in heat-maps as in Fig. 3a, d. An overview of the chromosomal positions of the investigated loci is shown in the top right cartoon. The data for Ori is the same as in Fig. 3a, d. Cyan lines indicate the predicted cell size of replication based on the model where initiation occurs at 2.05 µm², replication

progresses at a constant rate and takes 45 minutes to complete, and cells grow exponentially with a doubling time of 50 minutes. The experiments with locus IDs: Ori, Ter, R2-5, and L2-5 were performed twice, and the experiments with locus IDs: R1, L1, R5, and L5 were performed once. Repeated experiments with locus IDs: Ori, Ter, R2-5, and L2-5 are shown in Supplementary Fig. S6. Statistics can be found in Supplementary Dataset 1.

## Replication-induced locus repositioning persists across different growth conditions

The only previous study using time-lapse microscopy to study the spatial interactions between different loci and the replisome in cells carrying both locus and replisome labels was done by Reyes-Lamothe et al.[25]. Their study was performed in growth conditions where the cells grow slower than in the conditions used in Figs. 1–5. Based on previous

observations[29], the spatial organization of the replisome is kept at slower growth, but initiation occurs primarily on one instead of two *oriCs*. To test if the difference in growth condition affects the dynamic behavior of the chromosome in relation to the replisome, we imaged four selected strains in M9 glycerol (0.2%) minimal medium. In the slow-growth medium, the average generation time is ~130 min, as compared to 60 min in M9 succinate (0.4%) supplemented with amino

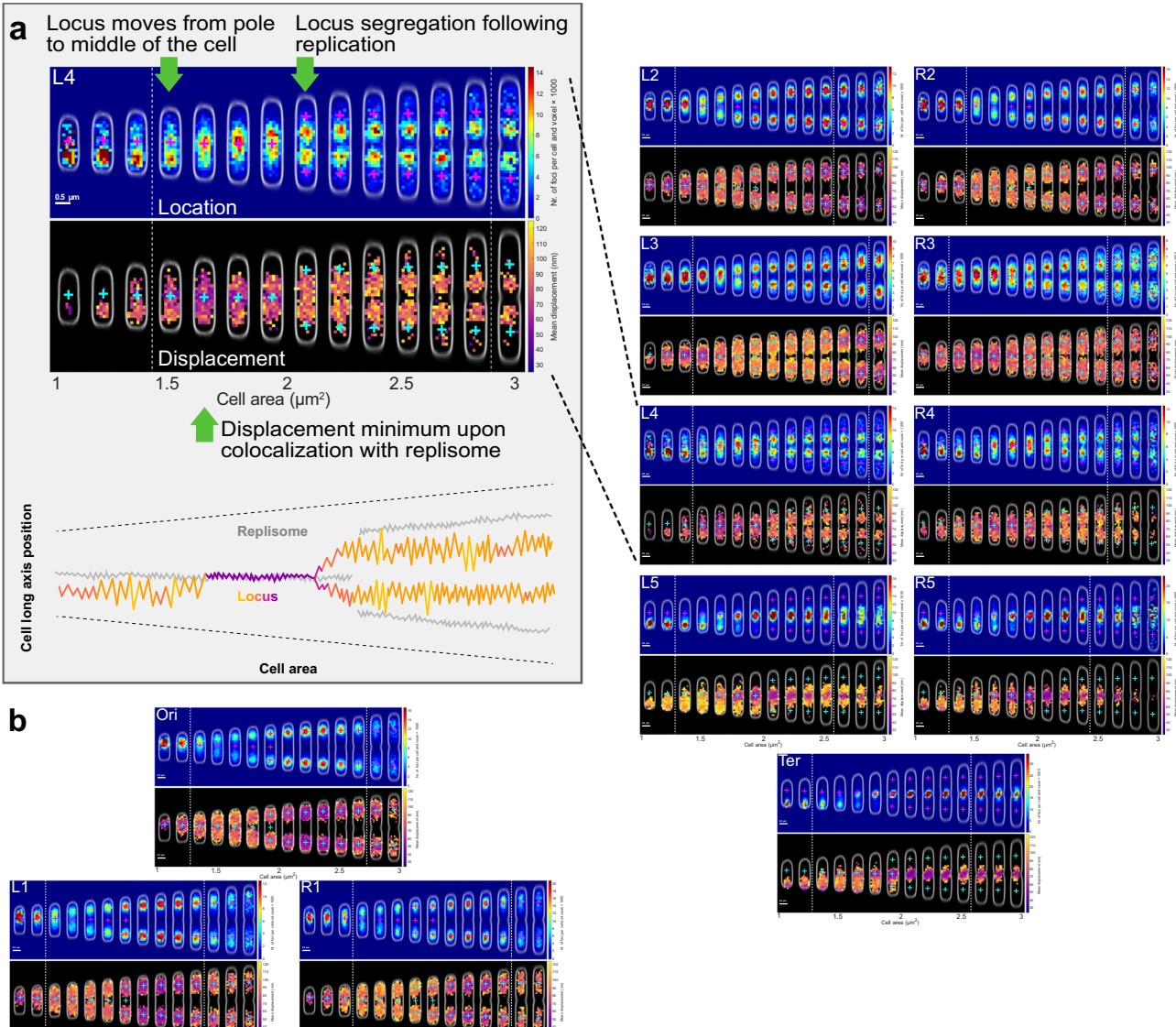

**Fig. 5 | Locus short-time-scale movement at different spatial positions.**
**a** *Location panel*: Two-dimensional histograms of fluorescent chromosome marker foci positions as in Fig. 1b but for locus L4 (see Fig. 4). Green arrows highlight locus relocation and segregation events. *Displacement panel*: Spatially binned loci displacements, where each bin shows the average length of 1-second displacements for which the first point is located inside the boundaries of the bin. The position and size of the bins are the same as in the location panel above. Each bin has a non-black color only if the number of displacements in the bin is larger than 4. Bins with fewer than five displacements are black. Magenta (in the location panel) and cyan (in the displacement panel) plus signs correspond to the mean position of the replisome distributions from the same set of cells used for the L4 location and displacement distributions. The green arrow highlights a cell area bin where a minimum in displacements (dark purple bins in the displacement panel) colocalizes with a high

density of replisomes (red bins in location panel). *Cartoon:* An interpretation of the location and displacement distributions above. Here, the positions along the cell-long axis of the replisomes are shown in gray. The position of the L4 loci is shown with the same colormap as in the displacement histogram above, ranging from purple, which indicates small 1-second displacement, to yellow, which indicates large 1-second displacements. **b** Location and displacement distributions as in a, for the 12 strains used in Fig. 4. The experiments with locus IDs: Ori, Ter, R2-5, and L2-5 were performed twice, and the experiments with locus IDs: R1, L1, R5, L5 were performed once. Repeated experiments with locus IDs: Ori, Ter, R2-5, and L2-5 are shown in Supplementary Fig. S7. Statistics can be found in Supplementary Dataset 1. Statistics of the number of cells and fluorescent foci can be found in Supplementary Dataset 2.

acids (Figs. 1–5). Also at a slower growth rate, the average short-time-scale movement of the replisome was lower than for the chromosome loci (Figs. S8 and S9). Similarly to the displacement results in Fig. 4, the displacement minima occurred upon colocalization with the replisome at different stages of the cell cycle for each locus (Fig. 4, Fig. S8). Similar to growth in M9 succinate, we found that loci L3 and L4 move towards the replisome to be replicated and that their replication was followed by segregation of the two loci copies (Fig. 6). In contrast to growth in M9 succinate with amino acids, however, both the L3 and the L4 locus location distributions could be split into two populations. One

population had the L3/L4 locus close to the old pole after division, while the other population had the L3/L4 locus positioned close to the new pole (Fig. 6). We will refer to this phenomenon as chromosome inversion. Similar inversion in the chromosome structure has also been observed previously[16,17].

We also repeated the experiments in a growth condition where the cells grow with a doubling time of 43 minutes. As expected[29]; changing from succinate to glucose in the medium primarily affected the cell size at birth, leaving the cell size at initiation relatively unaffected. The trend that loci short-time-scale movement decreases

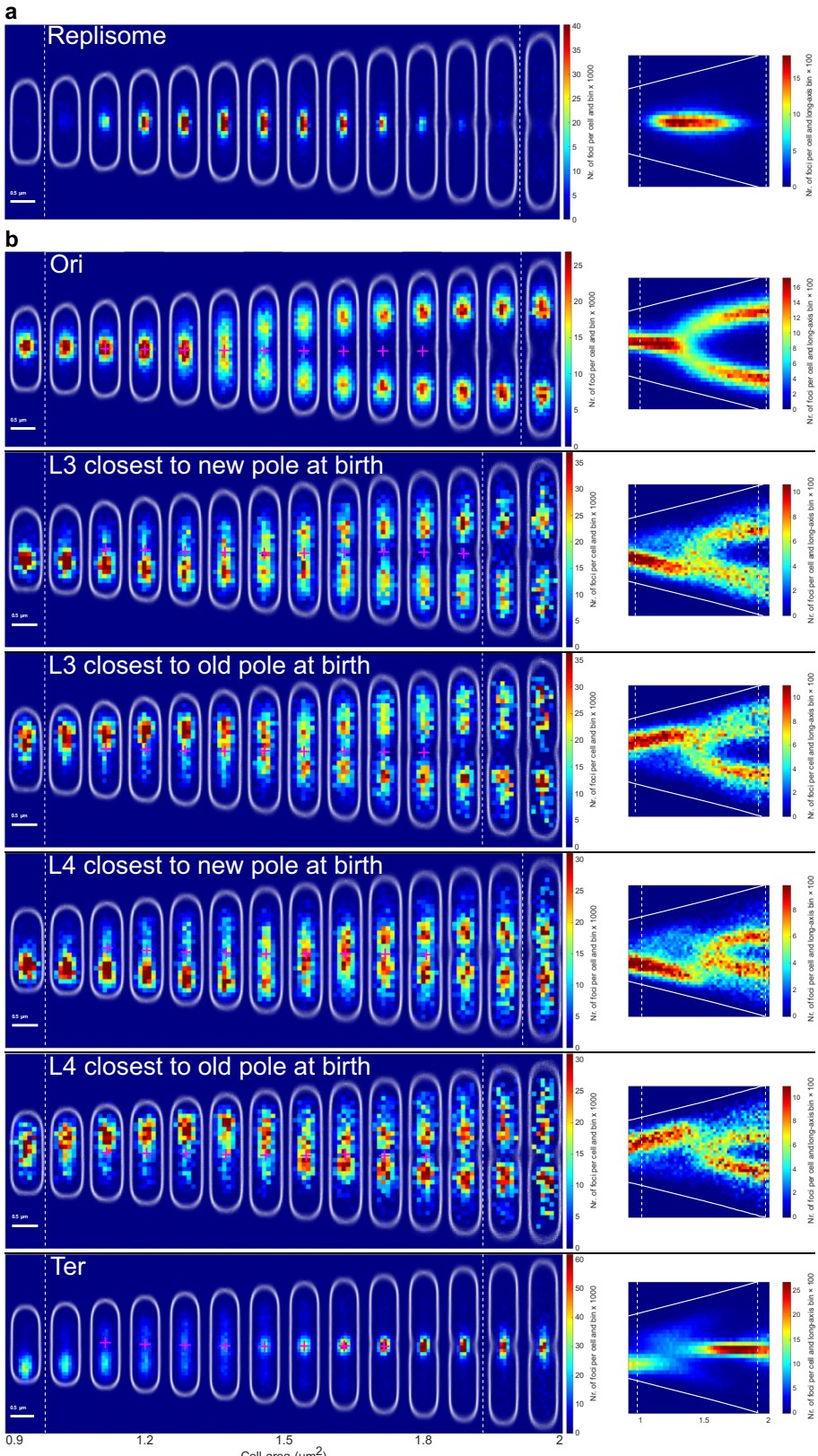

**Fig. 6 | Locus location distributions at slower growth rates.** Two-dimensional histograms of fluorescent foci positions along the long and short axes of the cell, as in Fig. 5, for strains with locus (**b**), and replisome labels (**a**) imaged at slower growth rates. Projections of the foci positions along the cell-long axis are shown in the panels to the right. Solid white lines indicate the average cell pole positions. These experiments were performed twice, and only data from one replicate is shown. Statistics can be found in Supplementary Dataset 1. Statistics of the number of cells and fluorescent foci in the two-dimensional histograms can be found in Supplementary Dataset 2.

(~3 μm²) as they relocated towards the replisome remained the same for the L3 locus (Fig. S10). Since initiation occurs at about the same cell size for cells grown in succinate and glucose, also the cell size at which L3 moved towards the replisome (at ~3 μm²) and the cell size of the subsequent separation of the two L3 loci (at ~1.8 μm²) is the same in both growth conditions (Fig. 5 and Fig. S10). Given the change in division size, the L3 separation occurred close to birth for cells grown in glucose (Fig. S10), but well after birth for cells grown in succinate (Fig. 5). For the terminus loci, the minimum in short-time-scale movement occurred at the cell middle, where the number of replisome foci per cell has already decreased to low values. We currently have no explanation regarding the underlying reason for this discrepancy.

Taken together, we find the movements of the chromosome in relation to the replisome to be similar for the fast, intermediate, and, for one of the two subpopulations, slow growth. For the second slow-growing subpopulation, the movements of the loci in relation to the replisome behave as expected from intermediate and fast growth, but with the mid-replichore loci at the opposite pole at birth.

## Discussion

We show direct evidence that chromosomal loci make long-range intracellular movements both directly before and directly after replication while the replisomes stay relatively stationary. This behavior is most clear for the L4 and R4 loci (Fig. 5a). These loci translocate, on average, more than half of the cell length in 30% of the average generation time (Fig. 5a). During the same period, the average movement of the replisomes is 5% of the cell length. This behavior is similar in cells with doubling times of 130 (Fig. 6), 60 (Fig. 5), and 40 (Fig. S10) minutes, although the loci travel shorter distances in the fastest growth conditions tested. In all three conditions tested, the *ter*-proximal loci exhibit similar pre-replicative movement as the R4 and L4 loci, but display no post-replicative segregation since their replication occurs close to the intracellular space where *ter* resides for a majority of the cell cycle. Both the intracellular position and rapid relocalization of *ter* have been reported previously[11], and it has also been suggested that this process is a direct consequence of replication[23]. Additionally, across all tested conditions, the *oriC*-proximal locus moves towards the replication region well before its replication. Typical behaviors of four selected loci in cells grown in slow and intermediate growth conditions are displayed in the model cartoons of Fig. 7 (Fig. 7a, 130 mins doubling time, Fig. 7b, 60 mins doubling time).

The overview of the chromosome loci and replisome movements throughout the cell-cycle shown in Fig. 7 is clearly different from what is presented in refs. 9,10,12,25,30. At the same time, the model fits nicely with the conclusions from refs. 26,27, although these studies did not track replisomes and chromosomal loci in the same cells.

In line with previous observations[27,29], we find that replisomes stay spatially confined throughout the cell cycle (Figs. 1 and 2). The two replisomes can, within the region, temporarily be detected as separate fluorescent foci (Fig. S11)[27,29], suggesting that bidirectionally moving replisomes do not form stable multi-replisome complexes while replicating[27,34]. The recent observations by Japaridze et al.[30] support this idea, since replisomes can be made to lose the spatial confinement and instead trail along the chromosome by perturbing the cell shape. Also the work of Reyes-Lamothe et al.[25] shows that replisomes have the possibility of separating over extended periods of time. However, Chen et al.[35] have suggested that the two bidirectionally moving replisomes are bound to each other during the first half of the replication period, and free from each other during the second half. On the other hand, the microscopy data presented in ref. 35 also fits well with transiently separating foci at random time points (Fig. S11)[27,29] and the bulk experiments in ref. 35 are based on cells in which the replication process is blocked on one arm which may affect the overall chromosome organization. Taken together, this suggests that while it is possible for the replisomes to trail along the chromosome arms, this is not

the observed behavior for our strain of wild-type-shaped cells, at steady-state growth in three different conditions. The reason that the replisomes remain confined to a region through which the chromosome moves when replicated has been suggested to be an emerging effect of replicating ring polymers constrained into a rod-shaped container[18].

Since the chromosome-replisome dynamics were shown to be dramatically changed by perturbing the cell's physiology[30], this could also provide insights into the discrepancy between our observations and those presented in refs. 9 and 10. As ref. 27 points out, the replisome location distributions that were based on snap-shots of steady-state populations binned on cell size shown in refs. 9 and 10 are very similar to the replisome position distributions presented by us and others[27–29]. However, they present a different interpretation of when replication initiation occurs. The discrepancies between our interpretations and that of refs. 9 and 10 appear in their usage of temperature-sensitive DnaA mutants to synchronize initiation of DNA replication, which was needed in experiments predating time-lapse experiments. The synchronization procedure changes the ratio between chromosome content and cell size. It also affects the potential to trigger new initiation events following the return to the permissive temperature. Due to the perturbation of the cell physiology, this may change the chromosome-replisome dynamics, and thus, a comparison with the steady-state behavior of unperturbed cells may be misleading.

In addition to the discrepancy of the large-scale movements of the chromosome and replisome, described above, our displacement measurements (Fig. 3) show the opposite relation between loci and replisome short-time-scale movement compared to the results reported in ref. 25. A potential explanation for this discrepancy could be differences in labeling strategy, as our FROS labels are based on shorter arrays (12 repeats of binding sites as compared to 240) and are thus expected to move faster in the short-time-scale regime.

Among the loci that we investigated, the *dif*-proximal locus exhibits a specifically distinct short-time-scale movement pattern over the cell cycle (Figs. 4 and 5, Locus ID: Ter, 47 kb from *dif*). The decreases in short-time-scale movement of other loci are temporary, i.e. occurring only during replication, while Ter remains relatively immobile at mid-cell from replication until cell division. The prolonged decrease in short-time-scale movements fits well with the previously reported minima in short-time-scale movements close to terminus[24] and can be rationalized based on known interactions in the terminus region. Replication termination results in catenated copies of *ter*[14,36], which are subsequently unlinked by the combined activity of FtsK and Topoisomerase IV[36,37]. The *ter* region has also been shown to be anchored to the septal ring at the cell middle based on an interaction dependent on MatP and ZapB[23]. The MatP-ZapB interaction has been suggested to stabilize *ter* at the septal ring, which could be reflected in its distinctive displacement patterns. The catenation and binding to the septal ring could contribute to the prolonged decrease in short-time-scale movement and narrow location distribution, compared to other loci.

When comparing loci dynamics in the slow (130 min doubling time) and intermediate (60 doubling time) growth conditions, the *dif*-proximal locus behavior is very similar. At birth, it resides near the new cell pole and relocates to the position where division will occur. While the dynamics of the *dif*-proximal loci are retained between the two growth conditions, the L3 and L4 loci display high cell-to-cell variability at slow growth conditions. Based on previous observations using loci markers of different colors in the same cell[16,17], we expect the L3 and L4 to never reside at the same pole as R3 and R4. Hence, at slow growth, the mid-replichore region can have chromosome-wide cell-to-cell variability without affecting the *ter*-region. Importantly, the positions of the replisomes are unaffected by the mid-replichore inversion. When visualizing the two populations of L3 and L4 in slow growth (Fig. 6), the replisome distributions are always analyzed using the same set of cells as for the loci. If the replisome positions were affected by the

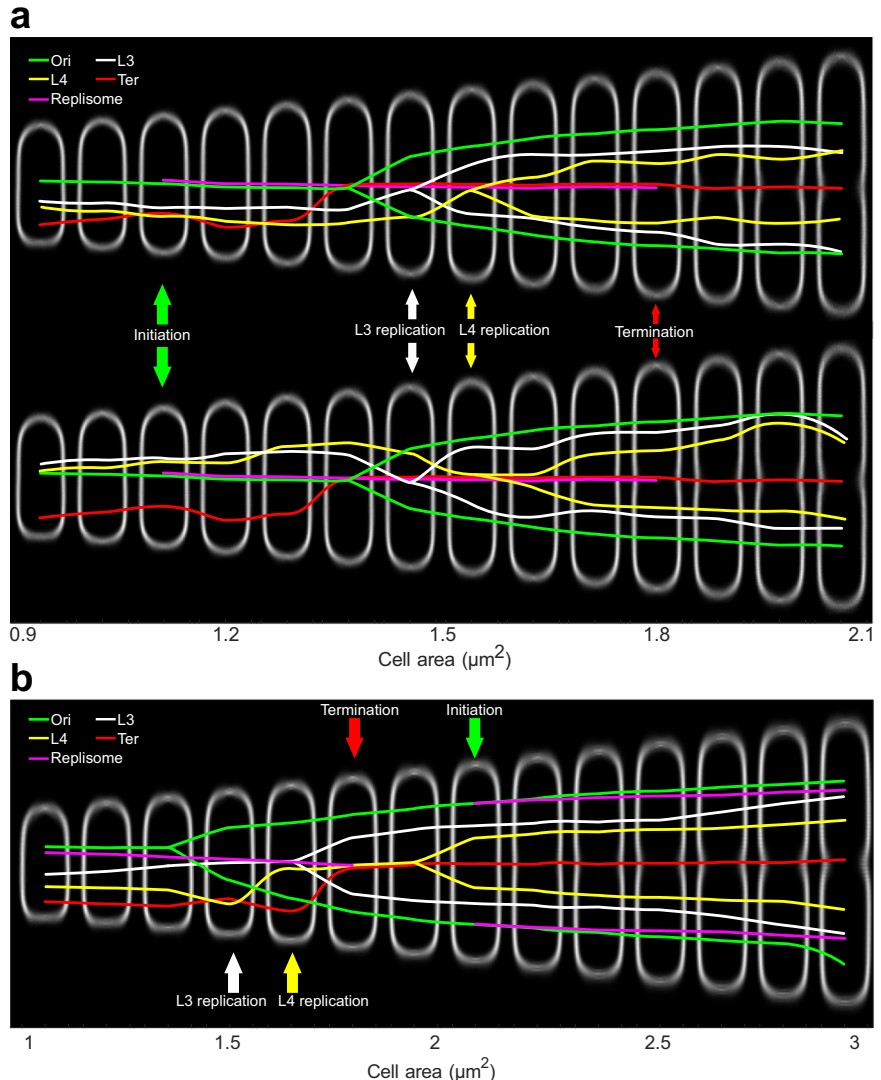

**Fig. 7 | Global rearrangements of locus and replisome positions over the cell cycle.** Cartoons illustrating the typical locations and replication of selected loci and replisomes in (**a**) slow and (**b**) intermediate growth. The two subpanels in (**a**) show the behavior of the two observed subpopulations, with L3/L4 found either closer to the new pole (top) or old pole (bottom).

mid-replichore inversion, we would, for example, expect the average location ('+' signs in Fig. 6) to be different in the two subpopulations. We also note some level of heredity[17] in the positions of the mid-replichore loci; the daughter cell corresponding to the cell half where the mid-replichore loci resided at birth in the mother have higher frequencies of receiving the inverted mid-replichore organization (Fig. 6).

It has previously been suggested that stationary bacterial replisomes could facilitate the segregation of newly replicated loci by extruding them towards the cell poles[6,38–40]. Replication as the main driver for chromosome segregation has been discounted due to the observation that loci do not form two distinguishable fluorescent foci immediately after replication[12]. Prolonged colocalization between locus copies following their replication has been described as a consequence of locus cohesion[21]. Defined as a tethering between replicated copies of a locus over a time period from locus replication to segregation[21], locus cohesion could contribute to the displacement minima in Figs. 3 and 4, as interlinked copies could be expected to diffuse slower than separated copies. As discussed in ref. 41, determining when a locus is segregated based on the localization of fluorescent foci alone is challenging, as two copies of a locus can be colocalized in a diffraction-limited focus. We note that, at intermediate

growth, the location distributions of *oriC*-proximal and *dif*-proximal loci split long after they were colocalized with the replisomes (Figs. 1 and 5b), while other loci exhibit a split in their location distributions soon after being replicated (e.g. L4, Fig. 5a). However, to conclude whether the displacement minimum is related to locus cohesion and segregation, we need an approach that does not rely on colocalization of foci that have the same color.

In this study, we have shown that *E. coli* chromosome loci generally exhibit larger short-time-scale movement than the replisomes. Replication takes place within a restricted region into which loci move when they are replicated. During the replication process, locus short-time-scale movement decreases and becomes more similar to the replisome's. We observe loci moving towards the replisome to be replicated and subsequently segregated in single cells. To conclude, these observations support a model with confined replisomes through which the chromosome moves to be replicated.

## Methods
### Strains and growth conditions
All constructed strains were derivatives of the strain *E. coli* MG1655 *rph + Δmall::frt ΔintC::P59-mall-SYFP2-frt ΔgtrA::SpR frt-mCherry-dnaN*. For

the construction of these strains, chromosomal modifications were performed using λ Red recombineering[42] and generalized transduction with phage P1 *vir* [43]. The strains and their genotypes are listed in Supplementary Table S1. Experiments at intermediate growth (Figs. 1–5 and 1–7) were performed in M9 minimal medium supplemented with 0.06× Pluronic F-108 (Sigma-Aldrich 542342) 0.4% succinate and 1× RPMI 1640 amino acid solution (Sigma). Experiments at slow growth (Fig. 6, Figs. S8 and S9) were performed in M9 minimal medium supplemented with 0.06× Pluronic F-108 (Sigma-Aldrich 542342) 0.2% glycerol. Experiments at fast growth (Fig. S8–10) were performed in M9 minimal medium supplemented with 51 μg/ml Pluronic F-108 (Sigma-Aldrich 542342) 0.4% succinate and 1× RPMI 1640 amino acid solution (Sigma). All experiments were performed at 30 °C. Strains were inoculated one day before each experiment in culture tubes with growth medium from frozen stock cultures stored at −80 °C. Cells were grown overnight at 30 °C in a shaking incubator (200 rpm). On the day of the experiment, the cells were diluted 1:100 in growth medium and grown for 3–4 h (intermediate growth) or 2–3 h (fast growth) before being loaded in the microfluidic chip. For experiments at slow growth, the cells were loaded from a culture grown overnight.

A mother-machine-type PDMS chip with open-ended channels[32] was used for all microfluidic experiments. The width and height of these channels was 1000 nm. Medium flow and loading of cells into the chip was performed by supplying pressure to the chip ports with a microfluidic flow controller built in-house (AnduinFlow). The imaging of the chip was performed in a H201-ENCLOSURE hood that enclosed the microscope stage, with the hood being connected to a H201-T-UNIT-BL temperature controller (OKOlab).

## Microscopy

All imaging was performed with a Ti-E (Nikon) microscope equipped with ×100 immersion oil objectives (Nikon, CFI Apo Lambda, NA 1.45, and Nikon, NA 1.45, CFI Plan Apochromat Lambda D MRD71970) that were set up for phase-contrast and widefield epi-fluorescence microscopy. Phase-contrast and fluorescence images were acquired of cells growing in a microfluidic chip at 30 °C in M9 minimal medium. The imaging was performed over 20 h (slow growth), 10 h (intermediate growth) or 15 h (fast growth), unless otherwise noted. Phase-contrast images were acquired once per position on the microfluidic device with a 1 min$^{-1}$ imaging frequency, except for experiments performed at fast growth where a 30 s$^{-1}$ imaging frequency was used. Fluorescence images were acquired once per position with a 1 min$^{-1}$ or 5 min$^{-1}$ imaging frequency. The results for 1 min$^{-1}$ (Figs. 1–5) and 5 min$^{-1}$ (Figs. S6 and S7) are the same, but faster imaging gives more data along with a modest growth rate impact. Each imaged position included 20 traps with ~200–300 cells in total per position.

For fluorescence image acquisition, the sample was irradiated with a 580 nm laser (VFL, MBP Communications) for mCherry excitation followed by a 515 nm laser (Fandango 150, Cobolt) for SYFP2 excitation. Both lasers were set to a 25 ms exposure time and 30 W/cm$^2$ power density at the image plane. The laser light was shuttered using an AOTFnC together with MDPS (AA Opto-Electronic) and reflected onto the sample using a FF444/521/608-Di01 (Semrock) triple-band dichroic mirror. Fluorescence images were acquired using a Kinetix sCMOS camera (Teledyne Photometrics). Imaging of the two fluorescence channels was performed back-to-back by two function generators (Tektronix) that triggered the lasers based on the camera acquisition. Emitted fluorescence was transmitted through a Bright-Line FF580-FDi02-T3 (Semrock) dichroic beamsplitter to separate the fluorescence for different channels. The separated fluorescence was then filtered through BrightLine FF01-505/119-25 (Semrock) and BrightLine FF02-641/75-25 (Semrock) filters and focused on two different areas of the sCMOS camera.

For the displacement measurements, the fluorescence imaging also involved an additional image acquisition in the two fluorescence channels at a given time interval after the first acquisition. The double acquisitions with a time interval were performed by using the SMART Streaming feature of the sCMOS camera, which allows for sequential acquisition of images with different exposure times. One of the acquisitions was used only to introduce the time interval between the two frames used for the displacement measurements.

Phase-contrast images were acquired with a 50 ms exposure time using a DMK 38UX304 camera (The Imaging Source). The light source used for phase-contrast was a 480 nm LED and a TLED+ (Sutter Instruments). The transmitted light was passed through the same FF444/521/608-Di01 (Semrock) triple-band dichroic mirror as the fluorescence and reflected onto the camera with a Di02-R514 (Semrock) dichroic mirror.

## Image analysis

The image analysis was performed using an automated image analysis pipeline that is primarily written in MATLAB R2022a (Mathworks) and previously described in ref. 44. Cell segmentation of phase-contrast images was performed using a nested Unet neural network[45]. Cell tracking was performed using the Baxter algorithm[46] and tracking of fluorescent foci was performed using the u-track algorithm[33]. Fluorescent foci of SYFP2 and mCherry were detected using a wavelet-based detection algorithm[47] and the detected coordinates were then refined using a maximum likelihood-based localization algorithm[48]. The localization was performed with the fitting of a 2D Normal distribution function to the fluorescent signal. To transform the localized focus coordinates between the cameras used to acquire fluorescence and phase-contrast images, landmark-based registration was performed of 500 nm beads (TetraSpeck, Thermo Fisher) that were visible on both cameras.

The estimation of replisome-*oriC* distances involved pairing of localized mCherry-DnaN and MalI-SYFP2 foci from the same cell. landmark-based registration was performed by imaging 100 nm fluorescent beads (TetraSpeck, Thermo Fisher) in two different fluorescence channels simultaneously on two different areas of the camera chip. The pairing between foci from two different fluorescence channels was performed based on this registration. Distances were estimated based on foci that were the closest to each other, and each focus was only paired with one other focus from the other fluorescence channel. The pairing of foci coordinates was performed using MATLAB's matchpairs() function.

Predicated replication sizes for the fluorescently labeled loci were calculated assuming exponential growth of the cells and using an average replication initiation size of 2.05 μm$^2$, an average generation time of 50 min, and an average C-period of 45 min. The replication size A($t$) is then estimated by $A(\alpha) = A_{Init} * \exp(\mu * C * \alpha)$, where $A_{Init}$ is the average replication initiation size, μ is the average growth rate, $C$ is the average C-period, and α is the genomic distance of a given locus from *oriC* divided by half of the length of the chromosome.

## Reporting summary

Further information on research design is available in the Nature Portfolio Reporting Summary linked to this article.

# Data availability

The raw microscopy data and analysis output generated in this study have been deposited in the Figshare database at https://doi.org/10.17044/scilifelab.25907971.

# Code availability

All code used for post-processing of analysis output and reproducing figures is available at https://doi.org/10.17044/scilifelab.25907971.

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

## Acknowledgements

We thank Irmeli Barkefors for the critical reading of the manuscript. Nynke Dekker provided us with strains containing the mCherry-DnaN construct. Elias Amselem built the two-color setup. This study was made possible by grants from the ERC (advanced grant no. 885360), the Swedish Research Council (grant nos. 2016-06213 and 2018-03958), the Knut and Alice Wallenberg Foundation (grant nos. 2017.0291, and 2019.0439) and eSSENCE. The computations and data management were enabled by resources provided by the Swedish National Infrastructure for Computing at UPPMAX, partially funded by the Swedish Research Council through grant agreement no. 2018-05973.

## Author contributions

D.F. and J.E. conceived the study; K.G., J.E. and D.F. designed experiments; K.G. constructed most strains, performed the experiments, and analyzed the data; K.G. and D.F. wrote most of the analysis code; K.G., J.E. and D.F. interpreted results; K.G., J.E. and D.F. wrote the paper.

## Funding

## Competing interests

The authors declare no competing interests.
