## [Peer Review File · Nature Communications]

The Escherichia coli chromosome moves to the replisomeReviewer #1 (Remarks to the Author):

The study of replisome localization has periodically resurfaced ever since the introduction of the "replication factory" model. This manuscript provides potentially powerful chromosome/replisome dynamics data, as opposed to most snapshot-based previous works. I strongly support its publication in Nature Communications, once the authors address several major presentational issues. Here are my detailed comments:

1. General Observations:

The manuscript offers insightful and potentially impactful data. However, in several sections, it appears a bit underwhelming given the extensive prior studies on this subject. It's imperative that the authors revisit their manuscript and the existing literature to pinpoint and elucidate the novel aspects of their findings.

(i) Section: 'Replication of a locus coincides with a decrease in its diffusivity'

- The results and Figure 3 need clarification. It's not evident how the data was collected and what it aims to showcase. The heatmaps across all panels need clearer labeling and interpretation.

- Is the data reflective of the steady-state population or is it specific to the initiation point?

Depending on this, the interpretation of the x-axis and the data concentration around $2-2.5\mu\text{m}^2$ can change significantly.

- If the study is showcasing the initiation point, why does the average initiation mass appear on the cloud's left-most periphery rather than its center?

- Shouldn't a stationary replisome, pulling DNA through for replication, display more directed motion of the labeled loci? The term 'diffusivity' in this context seems misplaced, especially when no distinct diffusivity analysis is provided apart from the displacement heatmaps.

- A reference standard for expected "diffusivity" would be beneficial to compare the observed values and justify 'high' vs. 'low' diffusivity.

- Why does the displacement increase $> 2.5\mu\text{m}^2$ so sharply?

- Due to the ambiguity in Figure 3, Figure 4 is also hard to follow (more on this below).

(ii) Section: 'Replication induces transient spatial repositioning of chromosomal loci'

- For Figure 4, integrating a loci map, similar to works by Stuart Austin (Genes Dev 2014) or Paul Wiggins (PNAS 2010), would be helpful for the reader.

- Why do the vertical dashed lines in the panels vary so significantly, given the sequential progression of replication from ori to ter and non-overlapping replication cycles?

(iii) Other General Observations:

- The concept of 'cohesion', as discussed by Hiraga, is noticeably absent. Can cohesion impact the interpretations of Figures 3 and 4?

- A robust model that aligns with the presented data would enhance the manuscript's depth.

Furthermore, predictions based on this model about chromosome replication and organization under various growth conditions could be invaluable.

- Could stationarity of replisomes be the result of the underlying chromosome structure (as discussed by Stuart Austin, Genes Dev, 2014, and other references thereafter)?

- More comprehensive explanations are needed to supplement the data. The manuscript would benefit immensely from visual illustrations or schematics to aid in clearer understanding.

Minor Points:

- For Figure 1, direct annotations would help, such as 'oriC (malO/'MalI)' on the upper panel and 'replisome (mCherry-DnaN)' on the lower one. Also, the vertical white & red dashed lines should be labeled directly on the panels.

Reviewer #2 (Remarks to the Author):

There appears to still be some debate as to whether E. coli replisomes are stationary or if they track along the DNA. This manuscript addresses this question using fluorescent labeling of DNA loci along with a marker for the replisome.

The data presented are indeed compelling and do show replisomes to be spatially confined arguing

against models put forth by references 10,11 and 12.

1) The data presented in Fig 3 and 4 needs a bit more explanation for the reader.

2) The portion of the paper about oriC localization to the membrane seems like an add on which does not flow with the rest of the manuscript. Although it seems clear that oriC is not a membrane bound locus like LacY the authors would need to add other experiments to bolster this section of the paper. What happens if SeqA is over expressed? Does oriC show more "membrane" localization under that condition? If oriC shows any true membrane localization, it would be expected to be very transient and not represent the dominant position of the origin.

3) There needs to be some discussion as to the rationale that the data provided here are somehow more reliable than the data presented in refs 10,11 and 12. What accounts for the differences that are observed by others.

Reviewer #3 (Remarks to the Author):

The ms of Gras et al describes the use of fluorescence imaging to study whether the E.coli replisome moves during chromosome replication.

There is conflicting literature regarding this question that dates back over 20 years. The two prevailing models are: the factory model whereby replisomes remain static and the DNA is translocated, and the tracking model whereby the replisome tracks over the DNA as it replicates.

The experiments in the ms are well executed and reasonably described. The results of OriC and replisome displacements at different cell areas are particularly original and support the notion that chromosomes translocate through stationary replisomes. However, the presentation of figures could be considerably improved (see remarks below), and additional analysis would be expected to support the main conclusions of the manuscript.

There are several lines of evidence supporting tracking and factory models in E. coli. The most recent studies using fluorescence imaging are from the Wiggins (Mangiameli, PLoS Genetics, 2017) and the Dekker labs (Japarizde, Nat. Comm. 2020). The present study provides additional support for the factory model, but the results and methods are similar to those already presented in previous studies (e.g. Mangiameli, 2017). The replisome localization over the cell cycle was reported using similar methods by Mangiameli et al (2017). Other results, such as demographs for FROS inserted at many different chromosomal positions were previously reported (e.g. Espeli lab). In fact, the tracking/factory model is perhaps an oversimplification, as replisomes were reported to split to remain together depending on growth conditions, and to perform excursions from one model (factory) to the other (tracking) during replication (Chen, Elife, 2023).

It is unclear whether the current study can truly exclude a possible combination of stationary and tracking models, as this does not seem to be addressed. Ultimately, because of these shortcomings, the ms does not provide enough additional evidence prompting a rethink of current models.

Lastly, we would recommend rewriting some sections of the manuscript for clarity. For instance, the "Initiation of replication does not occur at the cell membrane" section starts with a long paragraph describing the literature. We would expect this to be in the introduction of the manuscript, not within the results, as we have the feeling we are reading a different paper. This highlights also the fact that this section seems actually to address a different question than previous sections and how these are related is unclear.

Additional major issues

Additional analysis or experiments would be required to provide evidence for some of the claims

made in the manuscript.:

The authors claim the oriC-proximal chromosome locus shows more dynamics over the cell cycle as compared to the replisome, however this does not seem to be shown. They show the RMSD of the replisome over 20 minutes but do not compare this result to the RMSD for the OriC locus.

For instance "At cell sizes of $\sim 1.6 \mu\text{m}^2$, the loci relocate from the old pole into the region of the replisome, in which the locus diffusivity becomes slower". We don't see in Fig 5 the decrease in diffusivity the authors refer to.

"indicating that the two replisomes moving bidirectionally on the chromosome stay close together throughout the replication process". We could not find proof that this is the case from your data, we surely missed it. Perhaps it all depends on the definition of 'close together', as their spatial resolution seems to be the diffraction limit.

As a follow up, the standard deviations in Fig. S1 shows that replisomes can potentially explore a region of 200nm around the 3/4 or midcell positions. Thus, the two replisomes could be this distance apart and not necessarily 'close together'. It is surprising that the authors do not attempt to track the two independent replisomes. This has been achieved before (see Mangiameli, 2017) and would be necessary in this ms to strengthen the authors conclusions and to provide additional inklink.

"These observations show that the chromosome moves as a result of the replication process and not vice versa." This is a strong statement and we don't see the evidence for it in their data.

"Contrary to the membrane involvement suggested by previous studies, we did not observe localization distributions reminiscent of membrane binding at any point in the cell cycle (Fig. 1)". To what extent the authors can support such a strong statement (contradicting well documented studies) from a histogram? One can easily imagine a short-lived excursion of the oriC to the membrane region that is not captured in a histogram just because it is short lived and therefore far less likely than a localization close to the long axis. Unless further experiments or analyses are presented, we do not think this represents enough support for this conclusion.

How can the distribution of a membrane protein, such as LacY be expected to be a model for what to expect from the replisome, that would briefly contact the membrane-proximal region upon replication initiation?

Some statements would profit from some toning down. For instance in the sentence: "We show striking evidence that chromosomal loci make large intracellular movements both directly before and directly after replication while the replisomes stay relatively stationary", the authors could have described the fact the stationary replisomes were reported before, and that the choreography of genomic loci is well established. These two data together support a model whereby the chromosome translocates through more or less static replisomes. In this, they provide additional evidence that diffusivity of genomic loci decreases during their replication, which provides additional support to the aforementioned model.

We would recommend verifying some of their statements, for instance "As the replisome and ter colocalize, replication terminates, which results in catenated copies of ter". If we remember well, XerCD recombination to decatenate chromosomes is only required in a small number of dividing cells (10-15%?). Their statement makes one think that this happens every division cycle.

Figures

Raw images are absent from the ms and would be reassuring to both reviewers and readers. Some panels are repeated several times. For instance it seems that Figure 1 appears again in Fig 6c and in Fig. 5 (top panel?). What is the need to show the same panels again?

Some figures are difficult to understand. For instance, we had a very hard time understanding what is being shown in Fig. 5. There are no labels on the many panels shown, nor a clear explanation in the text or the legends.

How many replisomes are detected per cell along the cell cycle? Do the authors ever detect splitting events in their conditions?

Statistics are lacking in all figures? How many cells? How many localizations? How many replicate experiments?

Fig1: clarity could be improved by indicating:

- Fluorescent label used
- Significance of white and red dash lines
- What does the colormap represent?
- How is the number of loci per cell normalized?
- Figures are generated from foci in 86930 cells → Give the individual number for each condition.
- How many replicate experiments were performed?

Fig S1:

Provide standard deviations on the mean STD values?

Provide N.

Display curves for Old and New poles with different colors?

Fig S1: →It is unclear what the authors are tracking. Are they tracking the mean position of the two replisomes or the position of individual replisomes at each replication focus?

Page 4: "due to its higher resolution compared to the mCherry-based label" → resolution or localization precision? If the authors meant resolution (we assume this because the point spread function of YPet is smaller than mCherry), could the authors explain why resolution matters here?

Fig2: the figure shows the RMSD of the replisome with the error bars corresponding to standard error of the mean. How many trajectories were used to compute these plots?

Fig S2:

Why is the short axis RMSD 50% larger than in the main figure while the long axis RMSDs are similar?

What are the statistics?

Page 4: "Taken together, we find that the localization distribution of the replisome varies less compared to the oriC-proximal locus, both in terms of its average position and width of the distribution" → Where have the authors quantified the OriC proximal locus distributions? How can the authors be sure the qualitative difference in fluorescent foci distributions shown in Fig1 is not due to the localization precision of the fluorescent proteins used (mCherry vs YFP)?

Figs. 3,4,5 and 6: How many cells, loci and replicate experiments were performed in each condition?

Fig. 6: Ter is known to associate with membrane-associated proteins though the authors' method does not detect this. Could the method remain too limited and not detect transient or asynchronous events (see above)?

No time lapse experiments supporting stationary replisome and translocating chromosomes through replisomes are presented. While this is not critical it would be reassuring to see the authors confirming their results using methods employed by other conflicting studies. Figures are not formatted properly. They lack scalebars, colorbar legends, consistent font size, etc...

REVIEWER COMMENTS

Reviewer #1 (Remarks to the Author):

The study of replisome localization has periodically resurfaced ever since the introduction of the "replication factory" model. This manuscript provides potentially powerful chromosome/replisome dynamics data, as opposed to most snapshot-based previous works. I strongly support its publication in Nature Communications, once the authors address several major presentational issues. Here are my detailed comments:

1. General Observations:

The manuscript offers insightful and potentially impactful data. However, in several sections, it appears a bit underwhelming given the extensive prior studies on this subject. It's imperative that the authors revisit their manuscript and the existing literature to pinpoint and elucidate the novel aspects of their findings.

We have now rewritten the introduction in order to highlight how our work is different from previous publications.

(i) Section: 'Replication of a locus coincides with a decrease in its diffusivity'

- The results and Figure 3 need clarification. It's not evident how the data was collected and what it aims to showcase. The heatmaps across all panels need clearer labeling and interpretation.

The results in Fig. 3 and 4 have been clarified by further explaining what we aimed to measure and how this was performed. See lines 220-231.

- Is the data reflective of the steady-state population or is it specific to the initiation point? Depending on this, the interpretation of the x-axis and the data concentration around $2-2.5\mu\text{m}^2$ can change significantly.

The data is reflective of the steady-state population. The size of cells in the steady-state population is used for binning, which is also what is used as the x-axis in Fig. 1, 3, 4, 5 and 6.

- If the study is showcasing the initiation point, why does the average initiation mass appear on the cloud's left-most periphery rather than its center?

The average initiation area is indicated in Fig. 3 with the red dashed line. We detect initiation events based on tracking of the fluorescent replisome label, where the start of a trajectory is identified as a replication initiation event. Hence, the initiation events are detected using a different method than the ones used for the estimation of displacements (Fig. 3a and 3b) or replisome-oriC distances (Fig. 3c). Additionally, the expectation that the average initiation area would appear at the center of the oriC displacement minimum depends on the interpretation of what the width of the minimum in Fig. 3a means. If the width of the displacement minimum corresponds to the variation in replication events, we acknowledge that the average initiation event would be expected to be in the middle of the minimum. The fact that we observe the “cloud” to the right of the estimated initiation point, could indicate that the loci stay close to each other for a while after replication, for example due to what has been described as cohesion.

- Shouldn't a stationary replisome, pulling DNA through for replication, display more directed motion of the labeled loci? The term 'diffusivity' in this context seems misplaced, especially when no distinct diffusivity analysis is provided apart from the displacement heatmaps.

The directed motion is on the time scale of minutes (Fig. 1) and it would be very difficult to detect on the time scale of 1 s that we use for the displacement measurements. Regarding the use of the term 'diffusivity', we have now adjusted the wording to use short-time-scale movement, instead of diffusivity throughout the manuscript.

- A reference standard for expected "diffusivity" would be beneficial to compare the observed values and justify 'high' vs. 'low' diffusivity.

We would like to clarify that 'high' and 'low' diffusivity were meant as relative labels when comparing the diffusivity of oriC and the replisome, or the diffusivities of the respective labels at different cell areas. When comparing Fig. 3a (oriC) and 3b (replisome) we observe that the average diffusivity of oriC is higher compared to the diffusivity of the replisome. Also, within the scope of Fig. 3a, the displacements become shorter, implying that oriC diffusivity becomes slower around cell sizes of 2-2.5 μm^2 .

We have reformulated large parts of the section that is now called "Replication of a locus coincides with a decrease in its short-time-scale movement". We have also included a new Fig. 3d to clarify the difference in displacements that we observe on the 1 s time scale.

- Why does the displacement increase $> 2.5\mu\text{m}^2$ so sharply?

We have extended Fig. 3 to also include examples of the displacement distributions at two different cell sizes, both for the replisome and the oriC-proximal locus (Fig. 3d). The oriC-proximal displacement is transitioning from a replisome-like distribution, exemplified at 2.2 μm^2 , to a distribution where the peak value has increased by ~50% , exemplified at 1.5 μm^2 . Note that since the cells are growing at steady-state and divide binary, an example at 1.5 μm^2 is the same as an example at 3.0 μm^2 .

- Due to the ambiguity in Figure 3, Figure 4 is also hard to follow (more on this below).

(ii) Section: 'Replication induces transient spatial repositioning of chromosomal loci'

- For Figure 4, integrating a loci map, similar to works by Stuart Austin (Genes Dev 2014) or Paul Wiggins (PNAS 2010), would be helpful for the reader.

We have now modified Fig. 4 to include a map showing approximately where the locus labels have been introduced on the chromosome relative to *oriC*.

- Why do the vertical dashed lines in the panels vary so significantly, given the sequential progression of replication from *ori* to *ter* and non-overlapping replication cycles?

As pointed out by the reviewer the position of the cyan lines (i.e. the expected cell size of loci replication) are based on the model in which replication progresses at a constant speed from *oriC* to *dif*. The model is described in the main text, in the figure legend, and in the method section. What we failed to describe is that, when the modeled loci replication size is larger than the average cell division, it is divided by two before being visualized. This is due to binary fission at steady-state growth. For example, in the L3 panel of Fig. 4 the cyan line was drawn at $1.4 \mu\text{m}^2$, but since cells are kept at steady-state growth and dividing by binary fission, a line drawn at $1.4 \mu\text{m}^2$ has the same interpretation as being drawn at $2.8 \mu\text{m}^2$. We have extended Fig. 4 to include the expected cell size at replication for both the case close to cell birth and the case close to cell division for the chromosome loci where it is applicable.

(iii) Other General Observations:

- The concept of 'cohesion', as discussed by Hiraga, is noticeably absent. Can cohesion impact the interpretations of Figures 3 and 4?

Locus cohesion has been described as a physical link between replicated copies of a locus over a cohesion period before the locus copies segregate. It is possible that the displacement minima in Fig. 3 and 4 are a result of locus cohesion, where the interlinked copies diffuse slower than separated copies. However, this possibility does not change our observations of displacement minima at replication. Whether the width of the displacement minimum actually reflects a cohesion period is challenging to determine, as two copies of a fluorescently labeled locus can be colocalized in a

diffraction-limited spot also without a physical link. Even if the two copies would be resolvable when colocalized, we have not been able to think of an imaging experiment that could prove the colocalization to be a direct consequence of cohesion. This would, for example, require something along the lines of a FRET-type measurement with the different fluorescent probes binding on the different homologous strands.

We would like to note that in Fig. 5 the localization distributions and displacement distributions over different spatial positions in the cell can be used to get an estimate of when loci replicate and segregate. If we define the splitting of localization distributions as the segregation of two locus copies and the locus displacement minimum as the locus replication event, the period between the displacement minimum and the localization distribution split should correspond to the average cohesion period. Based on this, we note that for loci R2, L2, R3, L3, R4, L4, the cohesion period appears very short, segregation occurs following locus replication. For loci R5, L5 and Ter, we do not observe post-replicative segregation until cell division. The loci Ori, R1 and L1 exhibit average displacement minima around $2 \mu\text{m}^2$ and segregation after cell division, suggesting a cohesion period of approximately half the cell cycle. Thus, based on the results shown in Fig. 5, a period of locus cohesion appears to only be observable for the loci closest to *oriC*, and not the other chromosomal regions that we investigated.

We have extended the introduction and discussion to include the concept of cohesion. Please see lines 456-468

- A robust model that aligns with the presented data would enhance the manuscript's depth. Furthermore, predictions based on this model about chromosome replication and organization under various growth conditions could be invaluable.

We have now extensively modified Fig. 5 to highlight the movement of loci towards the replisome and also included illustrations of a model that summarizes our results (Fig. 7).

- Could stationarity of replisomes be the result of the underlying chromosome structure (as discussed by Stuart Austin, Genes Dev, 2014, and other references thereafter)?

It seems likely, given the similarities between the chromosome structure model presented by Stuart Austin (2014) and our results.

- More comprehensive explanations are needed to supplement the data. The manuscript would benefit immensely from visual illustrations or schematics to aid in clearer understanding.

We have now revised several sections related to Fig. 2, 3, 4 and 5 to clarify what is being shown and highlight the main results of those figures. We have also added illustrations to Fig. 2, 5 and a new Fig. 7 to visualize what we wanted to investigate and what our conclusions are.

Minor Points:

- For Figure 1, direct annotations would help, such as 'oriC (malO/'Mall)'' on the upper panel and 'replisome (mCherry-DnaN)' on the lower one. Also, the vertical white & red dashed lines should be labeled directly on the panels.

We have now updated Fig. 1 with headings indicating which fluorescent labels were used for the replisome and oriC, as well as labels for initiation and division areas in the panels.

Reviewer #2 (Remarks to the Author):

There appears to still be some debate as to whether *E. coli* replisomes are stationary or if they track along the DNA. This manuscript addresses this question using fluorescent labeling of DNA loci along with a marker for the replisome.

The data presented are indeed compelling and do show replisomes to be spatially confined arguing against models put forth by references 10,11 and 12.

1) The data presented in Fig 3 and 4 needs a bit more explanation for the reader.

We have now tried to clarify the results in Fig. 3 and 4 by further explaining what we aimed to measure and how this was performed on lines 220-231.

2) The portion of the paper about *oriC* localization to the membrane seems like an add on which does not flow with the rest of the manuscript. Although it seems clear that *oriC* is not a membrane bound locus like *LacY* the authors would need to add other experiments to bolster this section of the paper. What happens if *SeqA* is over expressed? Does *oriC* show more "membrane" localization under that condition? If *oriC* shows any true membrane localization, it would be expected to be very transient and not represent the dominant position of the origin.

We agree with the reviewer and have chosen to remove the section about replication initiation at the cell membrane to improve the flow of the main text of the manuscript. We have instead introduced results from additional growth rates to support the main claims of the manuscript.

3) There needs to be some discussion as to the rationale that the data provided here are somehow more reliable than the data presented in refs 10,11 and 12. What accounts for the differences that are observed by others.

Both the introduction and discussion has been extended to more clearly highlight similarities and differences compared to previous studies and potential sources of

discrepancies between our observations and that of references 10, 11, and 12.

Japaridze et al. (2020) (reference 10) show that replisomes can move away from each other in a cell with an altered geometry, suggesting that there is no physical link between the replisomes. The separation of the replisome pairs is discussed on lines 379-392. Our results are not in direct conflict with the results of Japaridze et al. We observe that the replisomes are localized in a confined region in the cell, not that they are bound to each other.

Our observations are contradictory to the conclusions presented in Reyes-Lamothe et al. (2008) (reference 11). On lines 411-416 we speculate that this may be due to differences in locus labeling strategies. Additionally, we also performed experiments to measure the localizations and displacements of locus labels Ori, L3, L4, Ter and replisome labels mCherry-DnaN and YPet-DnaN in M9 glycerol (0.2 %) growth medium at 30 degrees. These growth conditions resulted in approximately 130 min generation time for the strains with locus and replisome labels, with replication initiation occurring on average soon after cell division. These conditions are close to the conditions used by Reyes-Lamothe et al. We present the results of the experiments with slower growth rates in a new Fig. 6, and conclude that the difference as compared to Reyes-Lamothe et al. is not due to growth rate differences.

The localization patterns presented by Kongsuwan et al. (2002) (reference 12) are similar to the localization distributions for the replisome shown in Fig. 1 in the manuscript. However, while the localizations appear to be similar, our interpretation is different. We interpret each replisome localization distribution (at the cell middle or $\frac{1}{4}$ and $\frac{3}{4}$ cell long axis positions) shown in Fig. 1 as having two replication forks. As we discuss on lines 380-392, fluorescent replisome foci can sometimes separate, showing the presence of two replication forks, although we detect them as a single focus most of the time. This is also the only interpretation that makes sense when we interpret DnaN foci over many different growth rates (Knöppel et al. 2023). In contrast, Kongsuwan et al. interpret the fluorescent foci as single replication forks, with the foci at the $\frac{1}{4}$ and $\frac{3}{4}$

long axis positions being described as separate forks moving away from each other. Kongsuwan et al. also note that due to photobleaching of their replisome label they were not able to confirm that the replisome focus forming at the cell middle separate into two replication forks.

Reviewer #3 (Remarks to the Author):

The ms of Gras et al describes the use of fluorescence imaging to study whether the E.coli replisome moves during chromosome replication.

There is conflicting literature regarding this question that dates back over 20 years. The two prevailing models are: the factory model whereby replisomes remain static and the DNA is translocated, and the tracking model whereby the replisome tracks over the DNA as it replicates.

The experiments in the ms are well executed and reasonably described. The results of OriC and replisome displacements at different cell areas are particularly original and support the notion that chromosomes translocate through stationary replisomes.

We are happy that the reviewer agrees with one of our main conclusions.

However, the presentation of figures could be considerably improved (see remarks below), and additional analysis would be expected to support the main conclusions of the manuscript.

We agree with the reviewer and have improved every figure in the manuscript. See more specific comments both above and below.

There are several lines of evidence supporting tracking and factory models in *E. coli*. The most recent studies using fluorescence imaging are from the Wiggins (Mangiameli, PLoS Genetics, 2017) and the Dekker labs (Japarizde, Nat. Comm. 2020). The present study provides additional support for the factory model, but the results and methods are similar to those already presented in previous studies (e.g. Mangiameli, 2017). The replisome localization over the cell cycle was reported using similar methods by Mangiameli et al (2017). Other results, such as demographs for FROS inserted at many different chromosomal positions were previously reported (e.g. Espeli lab).

We would like to highlight that although previous studies have used fluorescence imaging to study replisome or locus localization, we perform both in high-throughput time-lapse experiments. Having both replisome and locus labels in the same cells allowed us to connect the colocalization between them with locus displacement minima, which has not been described before. Large parts of the introduction were rewritten to more clearly highlight how our work extends beyond previous results. Please see lines 27-110.

In fact, the tracking/factory model is perhaps an oversimplification, as replisomes were reported to split to remain together depending on growth conditions, and to perform excursions from one model (factory) to the other (tracking) during replication (Chen, Elife, 2023).

Chen et al. (2023) clearly presents an interesting idea regarding the possibility of sister replisomes having the ability to cooperate in the fork progression when they are spatially proximal. On the other hand it provides no time-lapse data on intracellular spatial position of either chromosome loci or replisomes and can as such not give insight into the two models as defined by the reviewer:

“The two prevailing models are: the factory model whereby replisomes remain static and the DNA is translocated, and the tracking model whereby the replisome tracks over the DNA as it replicates.”

In the abstract of Chen et al. the two terms “factory configuration” and “solitary configuration” are used, which are more appropriate given the content of the paper. The factory model, as defined by the reviewer, is similar to the factory configuration shown in Chen et al, since it is hard to imagine that replisomes which cooperate in fork progression are actually tracking on the chromosome. It’s not completely unthinkable though, the only requirement is that the different arms of the chromosome lie side-by-side. The relation between the tracking model and the solitary configuration is more complicated. The requirement of the solitary configuration, as presented in Chen et al, is that two replisomes are resolvable as two fluorescent foci, which is possible already at say 200 nm (the exact distance will depend on the intensity and wavelength of the fluorescent foci, pixel sizes and parameters in the image analysis) and that the replisomes do not cooperate, as determined by bulk NGS data. This produces problems when applied to the two different models as defined by the reviewer. Here is why: The randomly separating foci, as observed in Mangiameli et al. (2017), which fits in the definition of the solitary configuration, are very different from the tracking model data presented in Reyes-Lamothe et al. (2008) where the replisome is continuously moving (followed in time-lapse microscopy) towards the chromosome loci, a total distance in the order of a μm . Hence, it is unclear where the solitary configuration fits into the current debate of whether the replisome is moving to the chromosome loci for the loci to be replicated, or whether the loci move to the replisomes to be replicated.

It is unclear whether the current study can truly exclude a possible combination of stationary and tracking models, as this does not seem to be addressed. Ultimately, because of these short-comings, the ms does not provide enough additional evidence prompting a rethink of current models.

If the solitary configuration as defined by Chen, et al is to be interpreted as being the same as the tracking model described in Bates and Kleckner (2005), Reyes-Lamothe et al. (2008) and Japaridze et al (2020) and the factory configuration of Chen et al is to be interpreted as the factory model as presented in Mangiameli et al. (2017) then the observations in our manuscript is in direct conflict with the observations in Chen et al.

Here is why: Chen et al. state that the replisomes transition from factory to tracking/solitary about halfway into the replication process. Based on this we would expect to see replisomes together at loci L1, L2, R1 and R2 and then they should start moving towards the chromosome loci at L3, L4, R3 and R5. This is not what we observe in Fig. 5, instead, the chromosome loci L3, L4, R4 and R5 are moving towards more stationary replisomes.

Lastly, we would recommend rewriting some sections of the manuscript for clarity. For instance, the “Initiation of replication does not occur at the cell membrane” section starts with a long paragraph describing the literature. We would expect this to be in the introduction of the manuscript, not within the results, as we have the feeling we are reading a different paper. This highlights also the fact that this section seems actually to address a different question than previous sections and how these are related is unclear.

We agree with the reviewer and have chosen to remove the section about replication initiation at the cell membrane to improve the flow of the main text of the manuscript. We have instead introduced results from additional growth rates to support the main claims of the manuscript.

Additional major issues

Additional analysis or experiments would be required to provide evidence for some of the claims made in the manuscript.:

The authors claim the oriC-proximal chromosome locus shows more dynamics over the cell cycle as compared to the replisome, however this does not seem to be shown. They show the RMSD of the replisome over 20 minutes but do not compare this result to the RMSD for the OriC locus.

We would like to clarify that in Fig. 1, we show that the oriC-proximal label moves far over the cell cycle, whereas the replisomes do not. The replisome localization distributions in Fig. 1 remain at the middle of the cell (before re-initiation of replication) or at the $\frac{1}{4}$ and $\frac{3}{4}$ cell long axis positions (at and after re-initiation). Over the same period of the cell cycle, two oriC localization distributions move continuously from the cell middle towards the $\frac{1}{4}$ and $\frac{3}{4}$ cell long axis positions (before re-initiation) and then keep moving towards the poles.

The RMSD of oriC over time relative to replication initiation is now included in a new Supplementary Fig. S2, which shows larger RMSD values compared to the replisome in Fig. 2. We have also extended Fig. 2 to include a cartoon illustrating the two cases that we wanted to differentiate by the experiment in 2b. As illustrated in Fig. 2a, we aimed to tackle the question of whether the width of the replisome localization distribution (Fig. 1, top panel) is a result of cell-to-cell variability or if it is a result of individual replisomes moving around. Hence, a comparison between the replisome and oriC RMSD is not directly relevant to answer the question. The text has also been updated to make the above distinctions more clear. Please see lines 180-196

For instance “At cell sizes of $\sim 1.6 \mu\text{m}^2$, the loci relocate from the old pole into the region of the replisome, in which the locus diffusivity becomes slower”. We don't see in Fig 5 the decrease in diffusivity the authors refer to.

Please see Figure R1 below where we have highlighted the events and the shifts in displacements that the above text was referring to.

Fig. R1: Long-range locus relocation towards the replisome. Two-dimensional histograms of fluorescent foci positions and displacements for locus L4, as in Fig. 5, with labels highlighting long-range locus movement towards the replisome and its replication.

“indicating that the two replisomes moving bidirectionally on the chromosome stay close together throughout the replication process”. We could not find proof that this is the case from your data, we surely missed it. Perhaps it all depends on the definition of ‘close together’, as their spatial resolution seems to be the diffraction limit.

We would like to clarify that this statement was in relation to the results shown in Fig. 1, top panel, where we observe that the replisome localization distributions at midcell or $\frac{1}{4}$ and $\frac{3}{4}$ long axis positions are relatively confined. Each localization distribution corresponds to two replication forks and we observe the two forks as a single fluorescent focus most of the time. Thus, the replication forks are localized in a region defined by the standard deviation of the 2D Gaussian fits of the localization distributions

shown in Fig. S1, which is what we intended to describe with the proximity of the replication forks. That being said, “close” can have different meanings for different readers as pointed out by the reviewer and we have now removed “close” and replaced it with the size of the region which the replisomes are confined within. Please see lines 172-178.

As a follow up, the standard deviations in Fig. S1 shows that replisomes can potentially explore a region of 200nm around the $\frac{3}{4}$ or midcell positions. Thus, the two replisomes could be this distance apart and not necessarily ‘close together’.

As described in the previous comment, we have reformulated our statements regarding the proximity of the replication forks to clarify that they explore a region that is defined by the standard deviation of the Gaussian fits of the localization distributions.

It is surprising that the authors do not attempt to track the two independent replisomes. This has been achieved before (see Mangiameli, 2017) and would be necessary in this ms to strengthen the authors conclusions and to provide additional inklink.

We certainly do. The algorithm used to link together one, or many, foci in subsequent frames of detection allows for both splitting and merging, much in the same manner as the algorithm used in Mangiameli 2017. The quantification of how often the two replisomes are observed as two separate foci is heavily dependent on image analysis parameters and foci are also easily lost if replisomes move out of the focal plane. In the answer to a similar question below we have included examples of separating foci.

“These observations show that the chromosome moves as a result of the replication process and not vice versa.” This is a strong statement and we don’t see the evidence for it in their data.

We agree with the reviewer that we cannot directly implicate the replication process and have updated the text to: “These observations are consistent with the model in which

the chromosome loci moves towards a region of spatially confined replisomes, in which the loci replicate. This is then also inconsistent with the model that the replisomes move to the chromosome loci before loci its replication.”.

“Contrary to the membrane involvement suggested by previous studies, we did not observe localization distributions reminiscent of membrane binding at any point in the cell cycle (Fig. 1)”. To what extent the authors can support such a strong statement (contradicting well documented studies) from a histogram? One can easily imagine a short-lived excursion of the oriC to the membrane region that is not captured in a histogram just because it is short lived and therefore far less likely than a localization close to the long axis. Unless further experiments or analyses are presented, we do not think this represents enough support for this conclusion.

We agree that transient membrane localization of the oriC would not be captured by a localization distribution such as the one shown in Fig. 1. We also mentioned this in the discussion of the previous version. Also, note that we increased the temporal resolution by binning the localization distributions of oriC and the replisome on time relative to the initiation event (Fig. 6c in the first submission). As stated above, the section about membrane-proximal initiations has now been removed from the manuscript, but we will keep this comment in mind and in a potential forthcoming publication we will quantify how short the short-lived interactions have to be in order to be lost in our analysis.

How can the distribution of a membrane protein, such as LacY be expected to be a model for what to expect from the replisome, that would briefly contact the membrane-proximal region upon replication initiation?

As described in the previous answer, the localization distributions we show in Fig. 1 will miss transient membrane localization events. We aimed to use LacY as a reference for the cell membrane, and compare its localization distribution to that of oriC and the

replisome to determine if membrane-proximal localization could be observed on the time scale of the cell cycle. Thus, LacY was not intended to be a model for what to expect from the replisome or oriC, only a more accurate reference for the cell membrane than what we get from the cell segmentation outlines. As stated above, the section about membrane-proximal initiations has been removed from the manuscript, but we will keep this comment in mind in potential forthcoming publications regarding the question about membrane-proximal initiations.

Some statements would profit from some toning down. For instance in the sentence: “We show striking evidence that chromosomal loci make large intracellular movements both directly before and directly after replication while the replisomes stay relatively stationary”, the authors could have described the fact the stationary replisomes were reported before, and that the choreography of genomic loci is well established. These two data together support a model whereby the chromosome translocates through more or less static replisomes. In this, they provide additional evidence that diffusivity of genomic loci decreases during their replication, which provides additional support to the aforementioned model.

The start of the discussion has been rephrased. See lines 356-357.

We would recommend verifying some of their statements, for instance “As the replisome and *ter* colocalize, replication terminates, which results in catenated copies of *ter*”. If we remember well, XerCD recombination to decatenate chromosomes is only required in a small number of dividing cells (10-15%?). Their statement makes one think that this happens every division cycle.

We would like to thank the reviewer for pointing out this mistake in the discussion. We have removed XerCD recombination from the discussion and now only highlight the interaction between *ter* and MatP, FtsK or ZapB as potentially contributing to the wider displacement minimum that we observe for our *ter*-proximal label.

Figures

Raw images are absent from the ms and would be reassuring to both reviewers and readers.

We have now added examples of tracked single-cells from fluorescence images in Fig. 1 as well as Supplementary Fig. 10.

Some panels are repeated several times. For instance it seems that Figure 1 appears again in Fig 6c and in Fig. 5 (top panel?). What is the need to show the same panels again?

The panels in Fig. 3 which are included in Fig. 4, and the panel in Fig. 1 which is included in Fig. 5 were repeated for easy comparison with the other panels in Fig. 4 and 5. This is also described in the figure legends.

Since the section about membrane-proximal initiation has been excluded from the manuscript Fig. 6 was also been removed. This being said, note that the results shown in Fig. 1 are not repeated in Fig. 6c. Fig. 1 includes replisome and oriC localization distributions sorted based on cell area. Fig. 6c includes replisome and oriC localization distributions sorted based on time relative to replication initiation.

Some figures are difficult to understand. For instance, we had a very hard time understanding what is being shown in Fig. 5. There are no labels on the many panels shown, nor a clear explanation in the text or the legends.

We have modified Fig. 5 to highlight the behavior we describe in the section "Replication induces transient spatial repositioning of chromosomal loci", and also extensively revised the sections that precede it to clarify what Fig. 3, 4 and 5 are showing.

How many replisomes are detected per cell along the cell cycle? Do the authors ever detect splitting events in their conditions?

Yes, we do detect splitting events. However, most of the time, pairs of replication forks appear as single foci in our conditions. Please see Fig. R2 (within this document) for examples of single-cell trajectories with the replisome label YPet-DnaN. In these trajectories we see cases of both prolonged and momentary replisome splitting events.

Fig. R2: Single-cell tracks of YPet-DnaN. Examples of single-cell tracks of a strain with the replisome label YPet-DnaN. Red arrows indicate cases of a single replisome focus splitting temporarily into two foci.

Statistics are lacking in all figures? How many cells? How many localizations? How many replicate experiments?

We thank the reviewer for pointing out that we failed to properly reference the statistics for our results. We have now included them in Table S2, which contains statistics for each figure, and we reference Table S2 in all figure legends.

Fig1: clarity could be improved by indicating:

- Fluorescent label used

We have now added headings with the fluorescent labels used to Fig. 1.

- Significance of white and red dash lines

We believe that removing the dashed lines from Fig. 1 would make the data more complicated to interpret. We find it difficult to orient ourselves in Fig. 1 and similar

two-dimensional histograms in the manuscript without knowing the average cell sizes at initiation and division, as we investigate events that occur over the division cycle or at the replication initiation event.

- What does the colormap represent?

In the figure legend of Fig. 1 we wrote “Color in each histogram heat-map indicates the number of foci per cell and histogram-voxel, where all the two-dimensional histograms use the same voxel-size and voxel positions.”. Please see lines 144-146. A legend has also been added to a colorbar in the figure summarizing this information.

- How is the number of loci per cell normalized?

We assume that *loci* is a typo. The number of *loci* cannot be directly observed from the fluorescence images, thus we present the number of fluorescent *foci* detected. Additionally, it is worth pointing out that none of the panels in Fig. 1 shows the number of foci per cell directly. The average number of foci per cell for each cell area bin can be found by summing up the value of each bin of the 2D histogram (i.e the heat-map). As stated in the response above, the number of foci per cell is what dictates the color in each histogram, there is no additional normalization.

- Figures are generated from foci in 86930 cells → Give the individual number for each condition.

We have now provided this information in Table S3.

- How many replicate experiments were performed?

We performed 2 replicate experiments for Fig. 1. This has now been specified in Table S2, along with statistics related to the 2 experiments.

Fig S1:

Provide standard deviations on the mean STD values?

See the replicate experiment in Fig. S3.

Provide N.

We have now included N in Table S2.

Display curves for Old and New poles with different colors?

Good suggestion! We have modified Fig. S1 to display standard deviations at the two cell poles in different colors.

Fig S1: →It is unclear what the authors are tracking. Are they tracking the mean position of the two replisomes or the position of individual replisomes at each replication focus?

We have updated Fig. S1. The results in Fig. S1b are based on 2D Normal distributions fits to the replisome localization distributions shown in Fig. S1a. We fit 2D Normal distributions multiplied by a constant to the heatmaps (2D histogram) of the replisome distribution and it is the width of the distributions, as defined by the std of the 2D Normal distribution, that is shown in Fig. S1. We have improved the description of how Fig. S1 was constructed both in the main text (lines 172-178) and in the figure legend of Fig. S1.

Page 4: “due to its higher resolution compared to the mCherry-based label” → resolution or localization precision? If the authors meant resolution (we assume this because the point spread function of YPet is smaller than mCherry), could the authors explain why resolution matters here?

We have now clarified that we used YPet-DnaN instead of mCherry-DnaN in the experiments for Fig. 2 as we find that YPet-DnaN has a higher signal-to-noise ratio on line 188. As we aimed to investigate whether the replisome localization distribution would arise from cell-to-cell variability or from the movement of individual replisome pairs, we chose to use YPet-DnaN as the replisome label to achieve better localizations and more reliable tracking of the replisome.

Fig2: the figure shows the RMSD of the replisome with the error bars corresponding to standard error of the mean. How many trajectories were used to compute these plots?

We have now provided the statistics for the experiment performed for Fig. 2 in Table S2.

Fig S2:

Why is the short axis RMSD 50% larger than in the main figure while the long axis RMSDs are similar?

As pointed out by the reviewer there is experiment-to-experiment variation in the RMSD estimates reported in the manuscript. Importantly, the conclusion made in the main text of the manuscript holds for both the RMSD curves in Fig. 2 and in Fig. S3. I.e. “The plateau values for both the short and long axis of the cell are similar to the expected average distance between two random positions in the replisome localization distributions, suggesting that the major contributor to the width of the replisome distributions shown in Fig. 1 is replisome movements and not cell-to-cell variation”

We have also included a cartoon in Fig. 2 which more clearly highlights the question that the experiment was aiming to answer.

What are the statistics?

We have now included statistics for the experiment performed for Fig. S2 in Table S2.

Page 4: “Taken together, we find that the localization distribution of the replisome varies less compared to the oriC-proximal locus, both in terms of its average position and width of the distribution” → Where have the authors quantified the OriC proximal locus distributions? How can the authors be sure the qualitative difference in fluorescent foci distributions shown in Fig1 is not due to the localization precision of the fluorescent proteins used (mCherry vs YFP)?

The quantified widths of the oriC-proximal distributions are now included in the supplement (Supplementary Fig. S2b). The fluorophores were assigned such that the fluorophore with the best signal (YFP) was used for the different chromosomal loci. The increase in localization precision when switching from mCherry to YFP is exemplified in the figure below where 1 second displacements are carried out in two different strains carrying either DnaN-mCherry (left) and DnaN-YFP (right). The DnaN-mCherry figure included here is the same as in Fig. 3b. The comparison below has also been included in a new Supplementary Fig. S4.

Figs. 3,4,5 and 6: How many cells, loci and replicate experiments were performed in each condition?

We have now included statistics of the experiments for Fig. 3, 4, 5 and 6 in Table S2.

Fig. 6: *Ter* is known to associate with membrane-associated proteins though the authors' method does not detect this. Could the method remain too limited and not detect transient or asynchronous events (see above)?

We are uncertain what the reviewer is referring to here. Close to cell division, terminus is localized in the middle of the cell, where the septum forms. Following cell division, terminus is found at the new cell pole, before relocating to the middle of the cell again, where it becomes replicated and remains there until the next cell division. Our results clearly show that terminus is localized at the septum near cell division (Fig. R3). Furthermore, terminus localization at the cell middle, where the septum forms, is neither transient nor asynchronous, it persists for almost half the cell cycle. The terminus localization patterns that we describe are similar to the localizations of cell division proteins ZapA, ZapB (Espeli et al. 2012) and MatP (Espeli et al. 2012, Sadhir & Murray, 2023), which act at the septum. In fact, our results are in absolute agreement with the model for terminus dynamics presented by (Espéli et al. 2012; Figure 7).

Fig. R3: The localization of *ter* at the cell membrane. Two-dimensional histograms in panels from Fig. 5 (Locus ID: *Ter*) with label highlighting the septal localization of *ter*.

No time lapse experiments supporting stationary replisome and translocating chromosomes through replisomes are presented. While this is not critical it would be reassuring to see the authors confirming their results using methods employed by other conflicting studies.

We would like to clarify that all experiments performed for the manuscript were time-lapse experiments, except for the ones including LacY-PAmCherry (previously Fig. 6, which is now removed from the manuscript). We have updated the main text to make this more clear. We have also modified Fig. 1 to include examples of single-cell tracks with fluorescence images of the replisome and oriC labels and added examples for the remaining loci in the Supplementary (Supplementary Fig. 10)

Figures are not formatted properly. They lack scalebars, colorbar legends, consistent font size, etc...

Scalebars have been added where appropriate.

Colorbars have been defined in the figures and in the figure legends.

Reviewer #1 (Remarks to the Author):

The revision is acceptable.

Reviewer #2 (Remarks to the Author):

The authors addressed my questions sufficiently. The introduction now reads like a review article as opposed to setting the stage for what the authors do and how they address the issue in the literature. The introduction should be revised.

Reviewer #3 (Remarks to the Author):

The authors have made an important effort in providing a literature overview on the debated model of replisome dynamics over the years and on clarifying these models in the introduction of the ms. However, there are multiple issues remaining that need to be solved.

In our previous review, we asked what we felt was a simple question: "...the tracking/factory model is perhaps an oversimplification, as replisomes were reported to split to remain together depending on growth conditions, and to perform excursions from one model (factory) to the other (tracking) during replication (Chen, Elife, 2023)"

The authors answer with a quite convoluted paragraph which is at times hard to follow. The data in the ms clearly supports a 'factory-like' model whereby replisomes remain spatially constrained and the DNA goes sequentially through this factory. It seems unclear how the solitary configuration fits into this model, and we would have expected a discussion of this in the ms.

In the PBP the authors make an attempt at explaining how their data would refute a solitary configuration model as defined by Chen et al. However, we were unable to understand the rationale of their answer given the lack of detail of what is being referred to in Fig. 5.

Line 162: "This shows that the two replisomes moving bidirectionally on the chromosome stay within a volume smaller than $\sim 125 \times 200$ nm ellipsoid". $\sim 125 \times 200$ nm makes it an area unit, not a volume. Also, it is unclear where this number was obtained from.

Line 172: "Since the replisome distributions in Fig. 1 showed a net average movement away from the cell middle on the time-scale of 20 min". Since there is no time indication of this figure it is difficult to know what net average movement the authors are referring to. On the figure we do not see any movement of the replisome from cell area $1 \mu\text{m}^2 \rightarrow 2 \mu\text{m}^2$ and similarly for replisomes at $\frac{1}{4}$ positions from cell area $2 \mu\text{m}^2 \rightarrow 3 \mu\text{m}^2$. In addition, we do not understand why this movement is subtracted since it is part of the replisome dynamics being quantified?

Line 180: The authors compare the RMSD of the replisome subtracted from their net average movement with the RMSD of OriC (we assumed the latter is not subtracted from their slow drift since there is no mention of it). How can the authors then compare the results from dynamics quantified differently?

Figures overall remain difficult to understand. Often data are split between the main figure and the supplement (e.g. Fig2) and this complicates comparison of relevant datasets. Often figures do not contain panels, complicating interpretation. The font sizes are at times unreadable. Colormaps lack legends. Choice of colors is confusing (e.g. Fig 3d). Additional comments on figures follow:

Figure 1:

Addition of panels would considerably aid readers in navigating the figure

Please indicate the colormap units (normalized fluorescence intensity level) on the new timelapse panels

The 'Time' label on these panels appears useless without any proper axis or units, and may be confusing if used with the same alignment and font size as "Birth" and "Division" labels.

Figure 2, panel a: Is the cartoon correct? Foci in the static heterogeneity section are still moving during the cell cycle, hence producing an MSD that cannot be flat. If this is the case, then we don't understand how from a RMSD curve in panel B they conclude dynamic heterogeneity if both dynamic and static heterogeneities would produce RMSDs that continue increasing with time.

Figure 5 is still very complex and hard to follow. This is unfortunate given that this seems to provide evidence against the solitary configuration model of Chen et al.

Minor points

Figure 2: why isn't the Fig S2 panel part of the main figure 2?

Figure 3 panel d should be swapped with panel c (introduced later in the text)

Figure 5 panels a and b may be swapped since panel b is introduced first in the main text.

REVIEWER COMMENTS

Reviewer #1 (Remarks to the Author):

The revision is acceptable.

Reviewer #2 (Remarks to the Author):

The authors addressed my questions sufficiently. The introduction now reads like a review article as opposed to setting the stage for what the authors do and how they address the issue in the literature. The introduction should be revised.

Reviewer #3 (Remarks to the Author):

The authors have made an important effort in providing a literature overview on the debated model of replisome dynamics over the years and on clarifying these models in the introduction of the ms. However, there are multiple issues remaining that need to be solved.

In our previous review, we asked what we felt was a simple question: "...the tracking/factory model is perhaps an oversimplification, as replisomes were reported to split to remain together depending on growth conditions, and to perform excursions from one model (factory) to the other (tracking) during replication (Chen, Elife, 2023)" The authors answer with a quite convoluted paragraph which is at times hard to follow. The data in the ms clearly supports a 'factory-like' model whereby replisomes remain spatially constrained and the DNA goes sequentially through this factory. It seems unclear how the solitary configuration fits into this model, and we would have expected a discussion of this in the ms.

In the PBP the authors make an attempt at explaining how their data would refute a solitary configuration model as defined by Chen et al. However, we were unable to understand the rationale of their answer given the lack of detail of what is being referred to in Fig. 5.

The discussion has now been extended with references to the Chen et al. data.

Line 162: "This shows that the two replisomes moving bidirectionally on the chromosome stay within a volume smaller than $\sim 125 \times 200$ nm ellipsoid". $\sim 125 \times 200$ nm

makes it an area unit, not a volume. Also, it is unclear where this number was obtained from.

The text was relying on the fact that the cells are rotationally symmetric in the short-axis direction. We have now updated the size to explicitly include the same distance in two directions. The text has also been updated on lines 161-163 to highlight that it is the standard deviation of the location distribution that is used as the measure of the width of the replisome region.

Line 172: "Since the replisome distributions in Fig. 1 showed a net average movement away from the cell middle on the time-scale of 20 min". Since there is no time indication of this figure it is difficult to know what net average movement the authors are referring to. On the figure we do not see any movement of the replisome from cell area $1\mu\text{m}^2 \rightarrow 2\mu\text{m}^2$ and similarly for replisomes at $\frac{1}{4}$ positions from cell area $2\mu\text{m}^2 \rightarrow 3\mu\text{m}^2$.

We have included a new Supplementary Fig. S4 that shows the average replisome movement due to growth that is subtracted for the RMSD estimation. As the reviewer points out, this movement is relatively small.

In addition, we do not understand why this movement is subtracted since it is part of the replisome dynamics being quantified?

To calculate the replisome region size we have all the replisome trajectories start from 0 at the time of replication initiation. This is done for the long and short axes, old and new poles separately. The trajectories are then followed for 20 min after the initiation event and we use this to estimate RMSD and investigate the size of the area covered by the trajectories over time. As the replisome region moves slightly due to growth, we subtract this movement to only investigate how large the replisome region is and not how it moves on average. If the average net movement of the entire replisome region is included in the RMSD it will no longer measure only the size of the replisome region, but instead the replisome region + the net average movement. Since we are only interested in the replisome region size in this specific experiment, the net average movement was subtracted.

Line 180: The authors compare the RMSD of the replisome subtracted from their net average movement with the RMSD of OriC (we assumed the latter is not subtracted from their slow drift since there is no mention of it). How can the authors then compare the results from dynamics quantified differently?

The drift subtraction was also performed for the RMSD estimation of oriC. It should have been stated in the main text, but was mistakenly omitted. The main text has been updated to include the drift subtraction for the RMSD of oriC in the figure legend of Fig. 2 on lines 196-197.

Figures overall remain difficult to understand. Often data are split between the main figure and the supplement (e.g. Fig2) and this complicates comparison of relevant datasets. Often figures do not contain panels, complicating interpretation. The font sizes are at times unreadable. Colormaps lack legends. Choice of colors is confusing (e.g. Fig 3d).

As noted below, we have added the oriC RMSD (previously Fig. S2) to Fig. 2 to facilitate a comparison of the replisome and oriC datasets without opening the supplement.

Panels have been introduced for data in Figs. 1, 4 and 5. The figure legend for Fig. 5 has been revised for clarity.

Colormap legends have now been added to the time-lapse panels of Fig. 1 and to the cell long-axis projections of Fig. 6.

Fonts have been updated but can, if needed, be further improved in collaboration with the editor.

Line legends for Fig. 3d have been updated.

Additional comments on figures follow:

Figure 1:

Addition of panels would considerably aid readers in navigating the figure

Please indicate the colormap units (normalized fluorescence intensity level) on the new timelapse panels

The 'Time' label on these panels appears useless without any proper axis or units, and may be confusing if used with the same alignment and font size as "Birth" and "Division" labels.

We have now added panels a, and b, to Fig. 1 and also updated the time-lapse panels and figure legend of Fig. 1.

Figure 2, panel a: Is the cartoon correct? Foci in the static heterogeneity section are still moving during the cell cycle, hence producing an MSD that cannot be flat. If this is the case, then we don't understand how from a RMSD curve in panel B they conclude dynamic heterogeneity if both dynamic and static heterogeneities would produce RMSDs that continue increasing with time.

Thanks for pointing out this inconsistency in the cartoon. Especially the green foci in the static heterogeneity panel above the "=" sign moved relative to the average. This was not our intention. Below the "=" sign the cartoon is correct and the foci do not move relative to the average position. The cartoon has now been corrected.

Figure 5 is still very complex and hard to follow. This is unfortunate given that this seems to provide evidence against the solitary configuration model of Chen et al.

Fig. 5 has been updated to include panels and the figure legend has been extended to further improve the readability.

Hopefully, this clarifies how our observations relate to all previous replisome train-track papers and not just Chen et al.

Minor points

Figure 2: why isn't the Fig S2 panel part of the main figure 2?

The oriC RMSD was not included in the main Fig. 2 because it is not relevant for determining if the replisome localization distribution is a result of cell-to-cell variability or replisome movement. We aimed to tackle the question of whether the width of the replisome localization distribution (Fig. 1, top panel) is a result of cell-to-cell variability or if it is a result of individual replisomes moving around. Hence, a comparison between the replisome and oriC RMSD is not directly relevant to answer the question.

This being said, different readers may have different opinions about distinctions between main text and supplemental data, and we have therefore also included the oriC data in Fig. 2.

Figure 3 panel d should be swapped with panel c (introduced later in the text)

We have updated the panel order in Fig. 3.

Figure 5 panels a and b may be swapped since panel b is introduced first in the main text.

We have updated the text to resolve this issue.